# A Fracture Rarely Comes Alone: Associations of Fractures and Stylolites in Analogue Outcrops Improve Borehole Image Interpretations of Fractured Carbonate Geothermal Reservoirs

Jasper Hupkes[1], Pierre-Olivier Bruna[1], Giovanni Bertotti[1], Myrthe Doesburg[1], and Andrea Moscariello[2]

[1]Department of Geoscience and Engineering, Delft University of Technology, Delft, The Netherlands
[2]Department of Earth Sciences, University of Geneva, Geneva, Switzerland

**Correspondence:** Jasper Hupkes (j.hupkes@tudelft.nl)

**Abstract.** In this study, we present a method that uses associations of discontinuity sets to demonstrate similarities between the outcrop and the subsurface. A discontinuity association comprises up to four discontinuity sets (fractures and stylolites) that can form coeval in a single stress field, a well-known concept that is rarely applied for subsurface characterization of discontinuities. We use this concept to improve the interpretation of borehole image logs of naturally fractured geothermal reservoirs in the Geneva Basin, Switzerland. Here, the naturally fractured Lower Cretaceous pre-foredeep carbonate rocks are targeted for geothermal exploitation, and exposures of this formation are found in three mountain ranges that surround the basin. In these outcrops, the orientations of the discontinuity associations are used as paleostress indicators in order to map out principal stress trajectories of regional discontinuity-forming events that created the background discontinuity network. We document two multiscale discontinuity-forming events that formed prior to Alpine fold-and-thrusting and thus constitute the regional-scale background network. Given the regional character of these events, we predict that the target reservoir is impacted by them as well. This prediction is subsequently used to isolate the background-related discontinuities on image logs from two boreholes that penetrate the target reservoir in the Geneva Basin. This analysis reveals that ∼45% of the observed discontinuities can be understood in the framework of the regional-scale background. In this way, we demonstrate that defining discontinuity associations in outcrops is a powerful tool to predict the geometry of natural discontinuity networks in the subsurface and subsequently can be used to develop geothermal exploitation strategies in naturally fractured reservoirs.

## 1 Introduction

Carbonate geothermal reservoirs with a low matrix porosity and permeability may still have a convective heat flow due to the presence of natural discontinuity networks (NDNs) (Berre et al., 2019; Medici et al., 2023). Natural discontinuities (fractures and stylolites) can create a heterogeneous reservoir permeability by either forming preferred flow pathways or barriers (e.g. Caine et al., 1996; Solano et al., 2011; Grare et al., 2018; Fadel et al., 2023). Also, natural discontinuities impact the rock strength, which is essential to consider in the case of hydraulic stimulation of a reservoir (Cao and Sharma, 2022; Rysak et al., 2022). Predicting the geometry of NDNs in the subsurface is therefore crucial to avoid production risks such as early thermal breakthrough (Fadel et al., 2023) and/or induced seismicity (Zang et al., 2014; Atkinson et al., 2020).

A large portion of the discontinuities are of sub-seismic scale. The only way to directly observe these discontinuities in the subsurface is through borehole data. These data classically consist of cores and borehole images. Borehole images (BHI) are a cost-effective alternative to core data, enabling classification of discontinuities based on their structural attitude (dip and strike) and their geophysical responses (filled — resistive or open — transmissive Williams and Johnson, 2004). However, there are several limitations to the usefulness of BHI-interpretations: for example, the resolution of BHIs inhibit identification of discontinuity type, and separating drilling-induced from natural fractures is not straight-forward (e.g. Lorenz and Cooper, 2017; Chatterjee and Mukherjee, 2023, and references therein). On top of that, it is challenging to place individual features observed on BHI into a broader context such as the reservoir, as different fracture histories may have resulted in the same fracture geometry observed at present-day (i.e., equifinality, see Laubach et al., 2019).

Because of the limitations of BHI data, outcrops are used as an additional source to characterize key attributes of the discontinuity network, such as length, spacing and connectivity in the subsurface (e.g. Agosta et al., 2010; Sanderson, 2016; Ukar et al., 2019). However, demonstrating that the outcrop and subsurface are similar is far from trivial (e.g Bauer et al., 2017; Peacock et al., 2022). To justify the usage of analogue exposures for characterizing discontinuities in the subsurface, not only similar lithology and age of formation is preferential, but also the diagenetic evolution and/or the evolution of mechanical properties (Bruna et al., 2019; Petit et al., 2022; Elliott et al., 2025) and tectonic history (e.g. Engelder, 1985; English, 2012; Petit et al., 2022) are important. For the latter, a comparison of the stress evolution of outcrop and subsurface is needed.

In outcrop studies, attempts have been made to group discontinuities based on the tectonic driver (e.g. Beaudoin et al., 2013; Aubert et al., 2019). Their formation is either related to local drivers such as folds and faults (Price, 1966; Torabi and Berg, 2011; Tavani et al., 2015) or to regional, far-field stresses (e.g., Bergbauer and Pollard, 2004; Casini et al., 2011; Lamarche et al., 2012; Bertotti et al., 2017; Lavenu and Lamarche, 2018; La Bruna et al., 2020). Discontinuities formed by the latter constitute the background network (sometimes called diffuse fractures). The distinction between drivers is relevant, as the spatial distribution and the intensity of the discontinuities is partially controlled by the driver. For local drivers, discontinuities concentrate where strain accumulates (e.g. near the fault damage zone or fold hinge), whereas the background network is present throughout the reservoir, with a spatial variability likely related to bed thickness, mechanical stratigraphy and diagenetic processes at time of fracturing (Bai and Pollard, 2000; Laubach et al., 2009; Barbier et al., 2012; Procter and Sanderson, 2018; Chemenda, 2022; Elliott et al., 2025). Understanding the genetic origin of discontinuities (i.e. background vs. fold/fault related) is therefore essential for extrapolation of discontinuity geometry to reservoir scale.

Stress fields in which discontinuities formed may be deployed to unravel the genetic origin of the discontinuity. There are various methods that use discontinuities for paleostress inversion, either based on slip vectors (e.g. Angelier, 1990; Maerten et al., 2016; Pascal, 2021), or roughness of stylolites (e.g. Beaudoin et al., 2016; Toussaint et al., 2018). Associations of discontinuities that formed coeval in a single stress field are more robust indicators than single discontinuity sets (see figure 2, Hancock, 1985). The concept that multiple discontinuities can form in a single stress field is largely sensed by structural geologist (e.g. Groshong, 1975), but surprisingly little used to establish the analogy between outcrop and the subsurface. This notion gives additional value to the surface study, as it provides a geological context for the interpretation of discontinuities in the target reservoir.

In this study, we use outcrops to predict the geometry of the background network in the subsurface of the Geneva Basin, Switzerland. We propose a methodology that utilizes associations of fractures and stylolites as the link between outcrop and subsurface reservoir. This concept is applied to the pre-foredeep Lower Cretaceous limestones. Recently, the Canton of Geneva supported different geothermal projects to exploit the subsurface of the Geneva Basin for cooling and heating applications (Geothermies https://www.geothermies.ch/, Heatstore https://www.heatstore.eu/). In the scope of these projects, two wells (GEo-01 and GEo-02) have been drilled in the basin for geothermal exploration of the Mesozoic carbonate rocks in the basin (Guglielmetti et al., 2021). Borehole data (including borehole images) of the two wells have shown that the Lower Cretaceous formations in particular are a potential geothermal reservoir, in spite of having a low primary porosity (<10%) and permeability (<0.1 mD) (Rusillon, 2017; Brentini, 2018; Moscariello, 2019; Clerc and Moscariello, 2020; Guglielmetti and Moscariello, 2021).

Based on the documentation of discontinuity associations in the outcrops, two events are defined that formed prior to tilting of the strata, and are consistently observed on different sides of the basin. The reconstructed stress trajectories of the two events are subsequently used to predict the geometry of the background network in the target reservoir. With this prediction, discontinuities related to the background network are identified on the borehole images of the two wells that penetrate the target reservoir. ∼45% of the discontinuities visible on the BHI fit within the predicted background network. This shows that grouping fractures and stylolites into associations provides valuable information for the characterization of the subsurface discontinuity network. The regional character of the background network can be used in future geothermal explorations in the Geneva Basin. Furthermore, the proposed methodology may serve as a guideline for any exploration project of a fractured geothermal reservoir.

## 2 Geological Background

The Geneva Basin is located in the western part of the Swiss Molasse Basin/North Alpine Foreland Basin (figure 1). It is bounded by the Jura Mountains in the northwest, the Salève Range in the southeast and the Vuache range in the southwest. In the subsurface of the basin, the Mesozoic strata are dipping 10° to 20° to the southeast. The depth of the top of the Lower Cretaceous increases from surface exposures in the northwest (Jura Mountains) to ∼1400 m in the southeast at the foot of the Salève Range (Jenny et al., 1995; Guglielmetti et al., 2020).

At present-day, exposures of the Lower Cretaceous are found in the mountain ranges surrounding the Geneva Basin and the Bornes Massif, part of the Sub-Alpine Chain (figure 1). In the Parmelan, part of the Bornes Massif, the Lower Cretaceous carbonates are exposed in the flat-lying plateau and steeply-dipping limbs of a box-fold (Berio et al., 2021). The largest exposure is on the plateau with a size of ∼2.5x2 km where the bedding is sub-horizontal and largely barren of vegetation. The box-fold is underlain by a NW-vergent thrust (Bellahsen et al., 2014) that structurally separates it from the Salève Range in the northwest. The Salève Range is also positioned above a NW-vergent thrust that marks the southeastern boundary of the Geneva Basin (Charollais et al., 2023). Here, the Lower Cretaceous is exposed in small outcrops (∼10 meters) on the SE-dipping limb of this range. On the other side of the Geneva basin, the Jura Mountains contains several Lower Cretaceous exposures, typically

in the river valleys, along road cuts and in abandoned quarries. The outcrop sizes vary between several meters to 100 meter. The mountain range is shaped by NW-verging folds and thrust formed by thin-skinned deformation (Homberg et al., 2002; Sommaruga et al., 2017). The Vuache Fault is part of a system of sinistral strike-slip faults related to the NW-vergent thrusting in the Jura (Homberg et al., 2002; Smeraglia et al., 2022). Transpression along this fault gave rise to the Vuache Range, with several exposures of the Lower Cretaceous carbonates along road cuts, with outcrop size up to ∼10 meters.

The Lower Cretaceous formations consists predominantly of carbonates, with intercalations of marl layers (Rusillon, 2017; Strasser et al., 2016). They are deposited in a branch of the Tethys Ocean (Clavel et al., 2007, 2013), and initially buried in the Late Cretaceous. The maximum depth during this first burial phase is poorly constrained, with estimates ranging from 800-1800 m in the Geneva Basin (Schegg and Leu, 1998), and up to 2000 m in the Sub-Alpine chain (Butler, 1991). Exhumation during the Paleocene led to sub-aerial exposure of the Lower Cretaceous carbonate rocks (Crampton and Allen, 1995). A second burial cycle placed the beneath Eocene to Pliocene molasse sediments in the foredeep of the Alpine Orogeny, whose thickness tapers towards the northwest. The estimated maximal burial depths of this second burial phase range between 2000 m in the Geneva Basin (Schegg and Leu, 1998) and 4000 m in the Bornes Massif (Butler, 1991; Moss, 1992; Deville and Sassi, 2006). In the Early Miocene, shortening in the Western Alps was accommodated by different folds and thrusts with a northwest vergence (Kalifi et al., 2021; Marro et al., 2023), ultimately leading to the exhumation of the Lower Cretaceous in the present-day mountain ranges in the Pliocene (Cederbom et al., 2004). Based on stress inversion of focal mechanisms of earthquakes in the Geneva Basin, the present-day stress field is pure strike-slip with $\sigma 1$ oriented NW-SE (Antunes et al., 2020).

## 3    Methodology

### 3.1    Grouping Discontinuities into Associations

The approach used in this study is based on outcrop observations of discontinuities, namely opening-mode fractures, small-scale faults, fracture arrays and stylolites, that are genetically associated with a certain stress field (figure 2). Discontinuity sets are defined on the basis of both orientation and discontinuity type (e.g. Lacombe et al., 2011; Sanderson et al., 2024). A theoretically complete discontinuity association (DA) consists of four sets of discontinuities: one opening-mode set, oriented perpendicular to $\sigma 3$; a conjugated pair of small-scale faults, with a 60° to 30° angle respectively, bisected by $\sigma 1$; and a stylolite forming perpendicular to $\sigma 1$. Relative timing based on cross-cutting and abutment relationships between individual sets within a discontinuity association is not considered in this study, as they do not reveal additional information on the orientation of the paleo principal stresses. In the study area, many fractures contain mineral deposits, and are then termed as veins. They are mainly found in arrays, belonging to semi-ductile shear zones that may form conjugate pairs similar to faults (Beach, 1975). As the aperture and infill of fractures in the outcrop do not necessarily coincide with subsurface (see e.g. Bauer et al., 2017), we use veins in the same way as other fractures to define DAs in the field. The consequences of this approach will be later discussed.

DAs are documented in the field per station with a size of around ten squared meters. Discontinuities that are compatible in a single stress field are thus observed in close vicinity of each other. In each station, we document the discontinuities that can

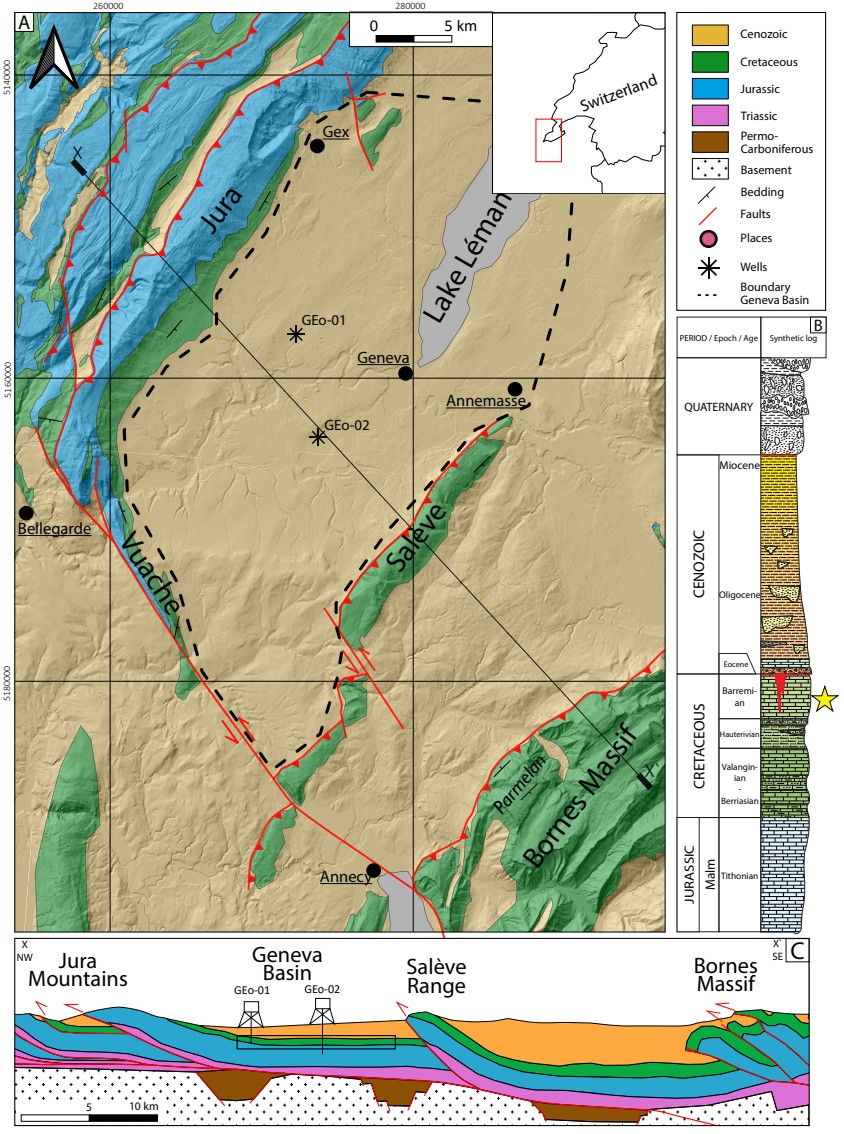

**Figure 1.** A) Simplified geological map modified after the 1:200.000 geological map of the Swiss Federal Office of Topography. The Geneva Basin (dashed black line) is located at the most western termination of the North Alpine Foreland Basin. Analogue outcrops of the Lower Cretaceous target reservoir are found in the Jura, Vuache, Salève and Bornes Massif (Parmelan) mountain ranges. Coordinates in UTM 32N reference frame. B) Synthetic log of the sedimentary succession in the Geneva Basin, from the Upper Jurassic upwards, after Moscariello (2019). The Malm and the Lower Cretaceous are both potential geothermal reservoirs in the basin. In this study, we focus on the Lower Cretaceous (marked with yellow star). C) Cross section showing the NW-verging Alpine thrusts that separate the target reservoir (black box) from the analogue outcrops. Modified after Bellahsen et al. (2014); Moscariello (2019); Kalifi et al. (2021); Marro et al. (2023)

be placed into a DA. The orientation of the DAs are used to map the related paleostress directions. To ensure the robustness of the reconstructed paleostress directions, we consider a minimum of two discontinuity sets that are associated together to define a DA. For example, a stylolite together with a conjugate pair of small-scale faults is considered a very reliable indicator. On the contrary, we discard features which provide ambiguous stress information, such as isolated opening fractures.

    This method inherently means that not all discontinuities observed are documented. To quantify how representative the de-

fined associations are for the total network, we measured augmented circular scanlines on seven pavements on the Parmelan (Mauldon et al., 2001; Watkins et al., 2015a). Per pavement, a total of four to twelve scanlines with a radius of one meter are collected. The orientation and type of discontinuities that intersect the circular scanlines are documented. The qualitatively defined associations from the nearest by station are used to separate the loose features that cannot be understood in the framework of an association from those that do. This gives an indication of the portion of discontinuities that fit within the framework of

associations with respect to the total network.

### 3.2    DAs in outcrop as prediction for the subsurface

The mapped paleo principals stress axes per station are used to determine the stress regime in which the DA was formed (i.e. normal, reverse, or strike-slip). We assume that all DAs are formed in Andersonian stress fields (Anderson, 1905), i.e. two of the principal stresses were positioned horizontally at the timing of discontinuity formation. This is used to reconstruct the

relative timing between the formation of a DA and the tilting of the strata (figure 2). If two of the three principle paleostresses are oriented parallel to the bedding, and the bedding is tilted, we infer that the DA formed prior to the tilting of the strata. Multiple DAs can be observed in a single station (in this study, we document a maximum of three). If the difference between the stress fields of two different DAs is only a permutation of the principle stresses, the simplest cause is a change in overburden (Bertotti et al., 2017) or by intermediate stress regimes (Simpson, 1997), rather than a different tectonic event. Therefore, these

DAs are grouped into single events.

    We use two criteria to define regional, background network forming events. Firstly, the relative timing of the DAs that make up the event must be prior to tilting of the strata. Secondly, the orientation of the principle stresses of the DA must be similar in all analogue outcrops that surround the target reservoir, i.e. constant on a regional scale. If the two criteria are fulfilled, we predict that the target reservoir that is located between the analogue outcrops, also is affected by these regional events. We use

this prediction to separate the background-related discontinuities observed on BHI of the two wells that penetrate the target reservoir, namely GEo-01 and GEo-02.

### 4    Results

We documented DAs at 28 different stations in the study area. Due to the quality of the exposure on the Parmelan, the majority (18) of the stations are documented there, with a total of three different DAs. The stations in the Jura and Vuache are described

together (9 in total). In these outcrops, a total of four different associations are defined. The third area is the Salève Range. Due

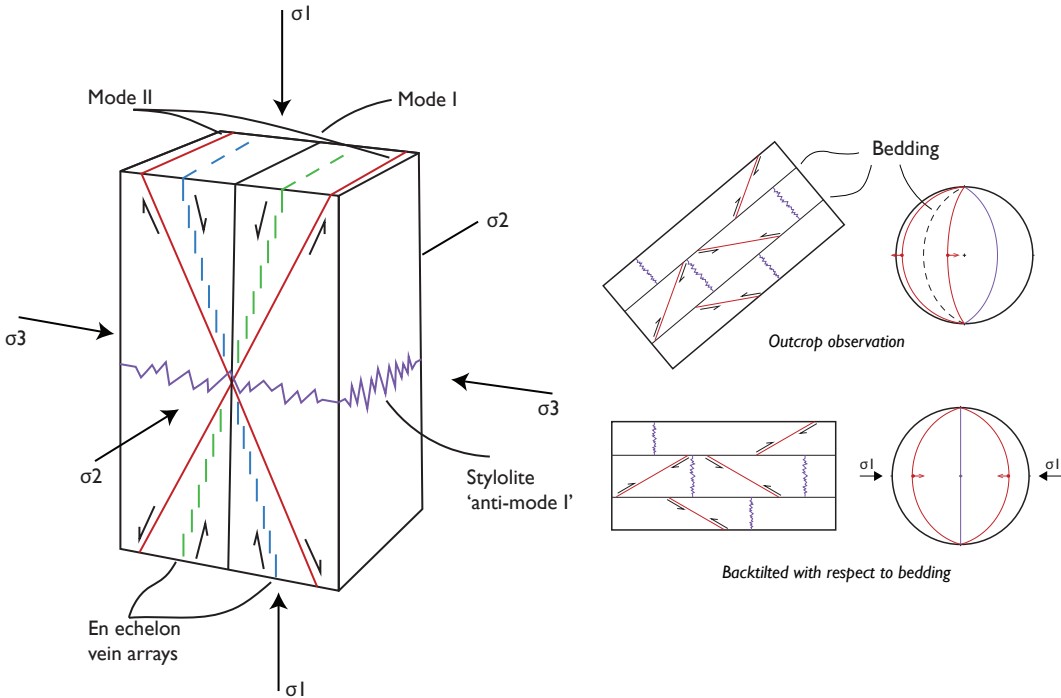

**Figure 2.** Left: Conceptual model illustrating a discontinuity association, adapted from Hancock (1985). There are four discontinuity sets that can form coeval in a single stress field, and are therefore in association with each other. For legend of colours, see figure 3. Right: illustration of how the timing of the formation with respect to tilting of the bedding is deduced. After back-tilting the association with respect to the bedding, the maximum principal stress becomes horizontal. Therefore, all the discontinuities that make the set (in this case, a conjugate pair of faults and a stylolite), formed prior to tilting of the strata and are thus part of the background network.

to limited exposures on top of this range, only one station is recorded here, where one association is defined. All the defined associations can ultimately be grouped into two regional discontinuity forming events that formed pre-tilting of the strata.

## 4.1 Parmelan

Veins, stylolites and small-scale faults are common on the Parmelan and can be arranged in discontinuity associations. The
160 oldest association (PA1) is expressed by a conjugate pair of small-scale faults and tectonic stylolites, observed on meter-scale. The faults strike ∼045°-225° with a low dip angle with respect to the bedding (∼15°-30°). Dissolution on the shear planes makes them clearly visible on bed-perpendicular exposures (figure 3A). In some instances, slickensides are preserved on the shear planes, indicating reverse kinematics. The tectonic stylolites have a similar strike as the small-scale faults, but are bed-perpendicular. The angular relationship between the faults, stylolites and bedding are observed everywhere, even in the steeply
dipping limbs of the box fold that shapes the Parmelan (e.g. station 18). Therefore, they are formed prior to the tilting of the strata. This association formed in a reverse stress regime with $\sigma 1$ oriented NW-SE.

The second association (PA2) comprises both small and large discontinuities, ranging from meter to kilometer in length. The smaller discontinuities are made up of two sets of sub-vertical vein arrays with opposing sense of shear (figure 3B) and tectonic stylolites. The latter similarly oriented as those of PA1. Sinistral arrays have an average strike of 150°-330°, whereas dextral arrays strike 120°-300° on average. The cement of the veins is yellow-white and have a blocky texture.

On a large scale, the plateau is dissected by two sets of fractures with similar orientation as the vein arrays (figure 4). They range in length from 100 to 3000 m. In the field, they appear as narrow bundles (<1 m) of smaller sub-vertical fractures. Dissolution along these fractures has created a karst system that is connected to an extensive cave system below the plateau (Lismonde, 1983; Masson, 1985). On the plateau, there are no offset markers that indicate horizontal displacement along these structures and no kinematic indicators are observed. At the northern edge of the plateau however, in front of the entrance to the Diau cave (see figure 10 for location), these fractures intersect a vertical cliff, and here bed-parallel slickensides on the fracture planes are preserved. The faults form a conjugate pair, so their structural attitude is similar to the vein arrays on top of the plateau, and are thus grouped into PA2. The associated stress regime of PA2 is a strike-slip regime, with $\sigma 1$ oriented NW-SE. This is similar to PA1, with the only change being a permutation of $\sigma 2$ and $\sigma 3$. Therefore, PA1 and PA2 together are considered as part of one event.

The third association (PA3) is also made up of a conjugated pair of sub-vertical vein arrays and tectonic stylolites, but with different orientations than those of PA2 (figure 3C). The dextral vein arrays strike ∼080°-260° and the sinistral arrays strike ∼120°-300°, with the cement of the veins being grey-coloured. The length of the discontinuities range from cm to 10s of meters. The stylolites are bed-perpendicular and strike ∼170°-350°. The veins that form the arrays occasionally cross-cut and displace those of PA2 (figure 3D), and are therefore interpreted as being younger in age. The paleostress regime related to this association is strike-slip, with $\sigma 1$ oriented ∼W-E.

To investigate how representative the above described associations are for the total network present on the Parmelan, we measured augmented circular scanlines on 7 different pavements on the Parmelan (see figure4 for location). Up to 75% of the total discontinuities observed can be understood within the framework of the predefined associations, and the majority belong to PA3 5. The percentages vary per pavement investigated, illustrating the spatial variability of the background network.

## 4.2 Jura and Vuache

Four different associations have been documented in the Jura and Vuache. In all cases, they formed prior to tilting of the strata. The first association (JA1) comprises a conjugate pair of small-scale faults with a low-angle with respect to the bedding and bed-perpendicular, tectonic stylolites (figure 6A). The strike of the small-scale faults and stylolites is ∼035°-215°. Slickensides on the shear planes indicate reverse kinematics. The reconstructed $\sigma 1$ of JA1 is oriented NW-SE.

The second association (JA2) is made up of a conjugated pair of bed-perpendicular small-scale faults (figure 6B), together with tectonic stylolites. Sinistral fractures strike ∼170°-350°, dextral fractures strike ∼110°-290°. Slickensides on the shear planes are always parallel the bedding. Tectonic stylolites strike of ∼035°-215°, similarly as those of JA1. The stress regime of JA2 is strike-slip, with $\sigma 1$ oriented NW-SE. The difference between JA1 and JA2 is a premutation of $\sigma 2$ and $\sigma 3$. Therefore, they are grouped into a single event.

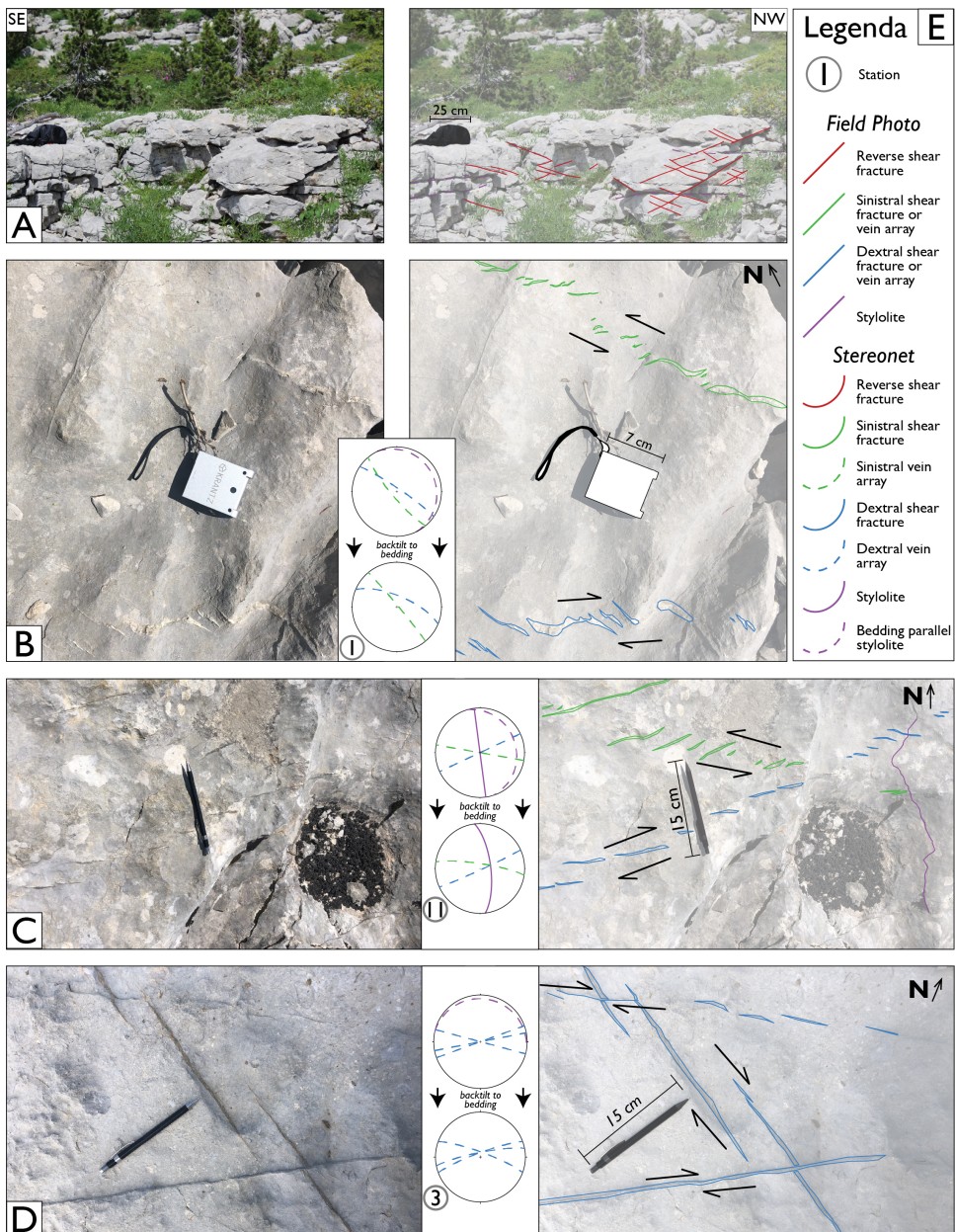

**Figure 3.** Field examples of DAs on the Parmelan. A) Low-angle conjugate pair of small-scale faults (PA1) with minimal displacement. Dissolution along the shear planes make them clearly visible on bed-perpendicular exposures. B) Top view image of a conjugate pair of vein arrays of PA2. C) Top view image of PA3 conjugate pair of vein arrays in association with a bed-perpendicular stylolite. D) Dextral vein array of PA3 crosscuts and slightly displaces a dextral vein array of PA2, indicating the relative timing of formation of the two associations. All stereonets, here in and in following figures, are lower hemisphere projections. E) legend for the colours used on the intepretation of the photographs and stereonets.

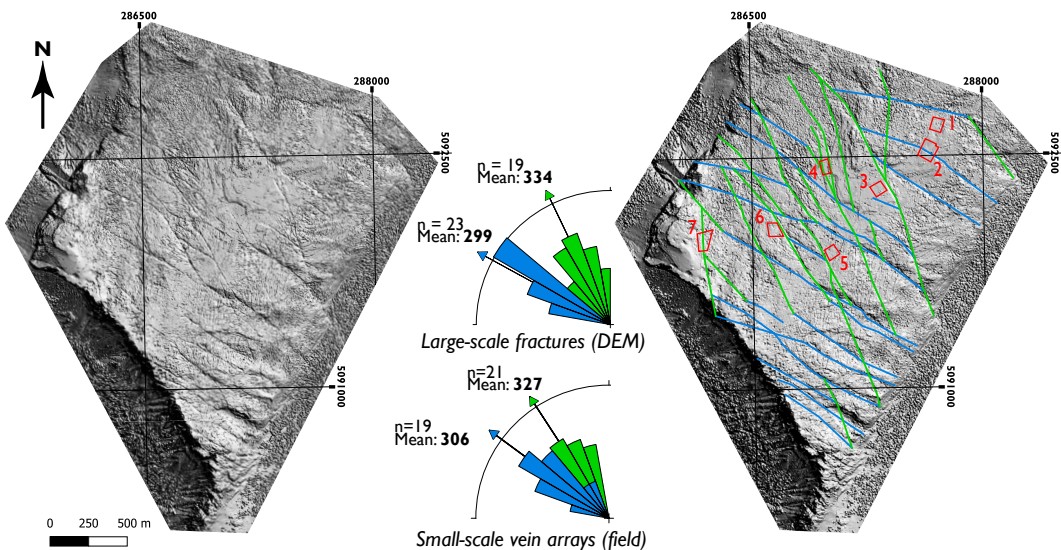

**Figure 4.** Digital elevation model of the Parmelan derived from LiDAR HD survey of IGN. The large-scale fracture network on the plateau is clearly visualized. The two main orientation of this network (upper rose diagram) correspond well with the orientations of the vein arrays of PA2 (lower rose diagram). At the northern plateau, opposing sense of shear is observed on these fractures, and therefore they are also grouped into PA2. The red insets refer to the pavements where circular scanlines are taken (see Table 1 for the results of the scanlines). Pavement 2 is shown as example in figure 5. For the colour code of the traced fractures, see legend of figure 3E.

The third association (JA3) is composed of ∼000°-180° striking reverse faults with a low angle with respect to the bedding (figure 6C), and bed-perpendicular tectonic stylolites. They formed in a reverse stress regime with $\sigma 1$ oriented W-E.

The fourth association (JA4)is expressed by bed-perpendicular ∼140°-320° striking sinistral faults and ∼075°-255° striking dextral faults (figure 6D), together with ∼010°-100° striking bed-perpendicular stylolites. This association is indicative of a strike-slip regime with $\sigma 1$ oriented W-E. This orientation is the same as JA3, and therefore they are part of the same event.

## 4.3 Salève

In the Salève, only one (SA1) is documented. Small-scale faults that strike ∼030°-210° and have a low angle with respect to the bedding (figure 7) are associated together with ∼035°-215° striking bed-perpendicular stylolites. The related stress field is a reverse regime with $\sigma 1$ oriented NW-SE.

## 4.4 Regional discontinuity-forming events

Based on the orientation of $\sigma 1$ of the associations in the different studied areas, we can define two regional discontinuity formation events (figure 9). The first event (E1) is characterized by a NW-SE trending, sub-horizontal $\sigma 1$. This orientation of $\sigma 1$ is both recorded by the reverse associations (PA1, JA1 & SA1) as well as by strike-slip associations (PA2 & JA2). In all outcrops, the reverse regime is similarly expressed by low-angle conjugate pairs of small-scale faults. Bed-perpendicular vein

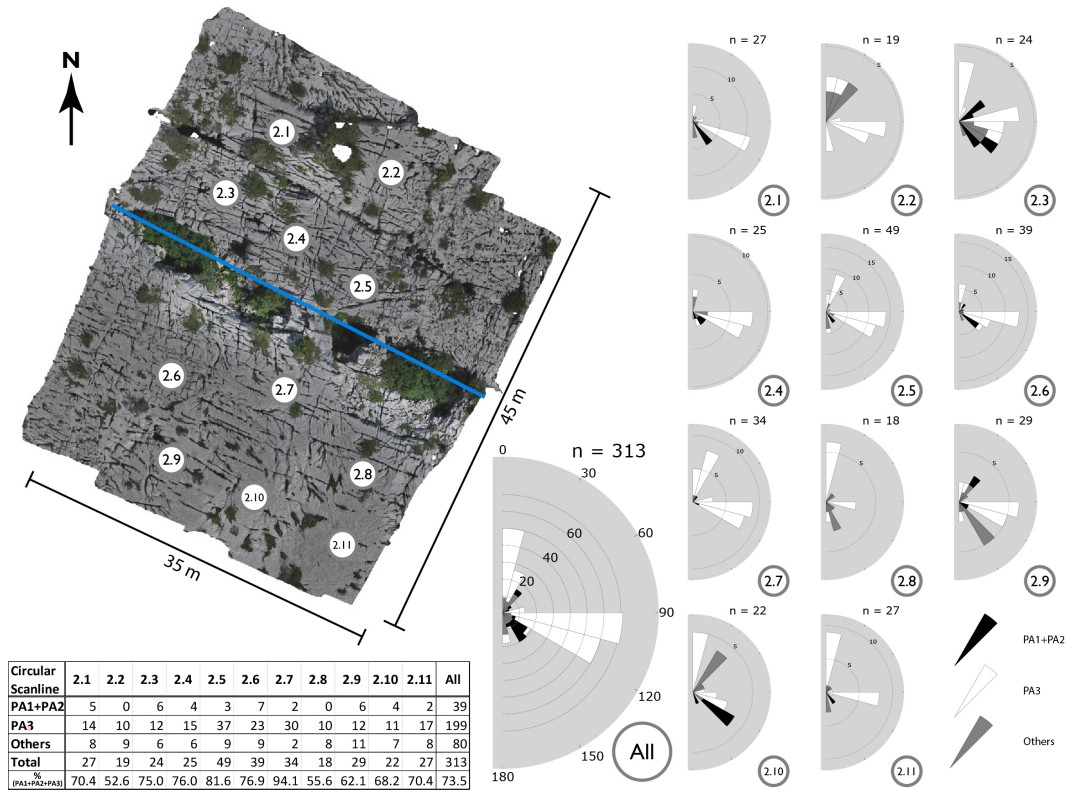

| Circular Scanline | 2.1 | 2.2 | 2.3 | 2.4 | 2.5 | 2.6 | 2.7 | 2.8 | 2.9 | 2.10 | 2.11 | All |
|---|---|---|---|---|---|---|---|---|---|---|---|---|
| PA1+PA2 | 5 | 0 | 6 | 4 | 3 | 7 | 2 | 0 | 6 | 4 | 2 | 39 |
| PA3 | 14 | 10 | 12 | 15 | 37 | 23 | 30 | 10 | 12 | 11 | 17 | 199 |
| Others | 8 | 9 | 6 | 6 | 9 | 9 | 2 | 8 | 11 | 7 | 8 | 80 |
| Total | 27 | 19 | 24 | 25 | 49 | 39 | 34 | 18 | 29 | 22 | 27 | 313 |
| % (PA1+PA2+PA3) | 70.4 | 52.6 | 75.0 | 76.0 | 81.6 | 76.9 | 94.1 | 55.6 | 62.1 | 68.2 | 70.4 | 73.5 |

**Figure 5.** UAV-derived orthorectified image of a pavement on the Parmelan (for location, see figure 4). Augmented circular scanlines with 1m radius show that the majority (75 %) of the observed discontinuities are in line with the qualitatively defined associations in station 4 (for location, see figure 10, for definition DAs, see figure 8 ). The scanlines are taken on both sides of a large-scale ENE-WSW fracture (blue line, see figure 4). There is no significant change in intensity of the discontinuities closer this fracture, suggesting that the large-scale fracture does not control the geometry of the total discontinuity network.

arrays of the strike-slip association are also observed in the Parmelan, Jura and Vuache. On top of this, in the Jura and Vuache, the strike-slip association is also expressed by conjugated brittle, sub-vertical faults.

The second event (E2) is also made up of a reverse and strike-slip association, but with a sub-horizontal $\sigma 1$ oriented ~W-E (figure9). In the Parmelan, only the strike-slip association is documented in the form of vein arrays and bed-perpendicular stylolites (PA3). In the Jura and Vuache, there is also a reverse association documented (JA3) with similar $\sigma 1$ as a strike-slip association (JA4). The latter is mainly depicted by bed-perpendicular faults, and less so by vein arrays, in contrast to the Parmelan.

A pre-tilting relative timing for E1 and E2 is observed in all studied areas. This implies that E1 and E2 were formed prior to Alpine fold-and-thrusting, and thus form the background network. As these events are consistently observed on all sides of the Geneva Basin where the target reservoir is located, we predict the presence of this background network in the subsurface as well.

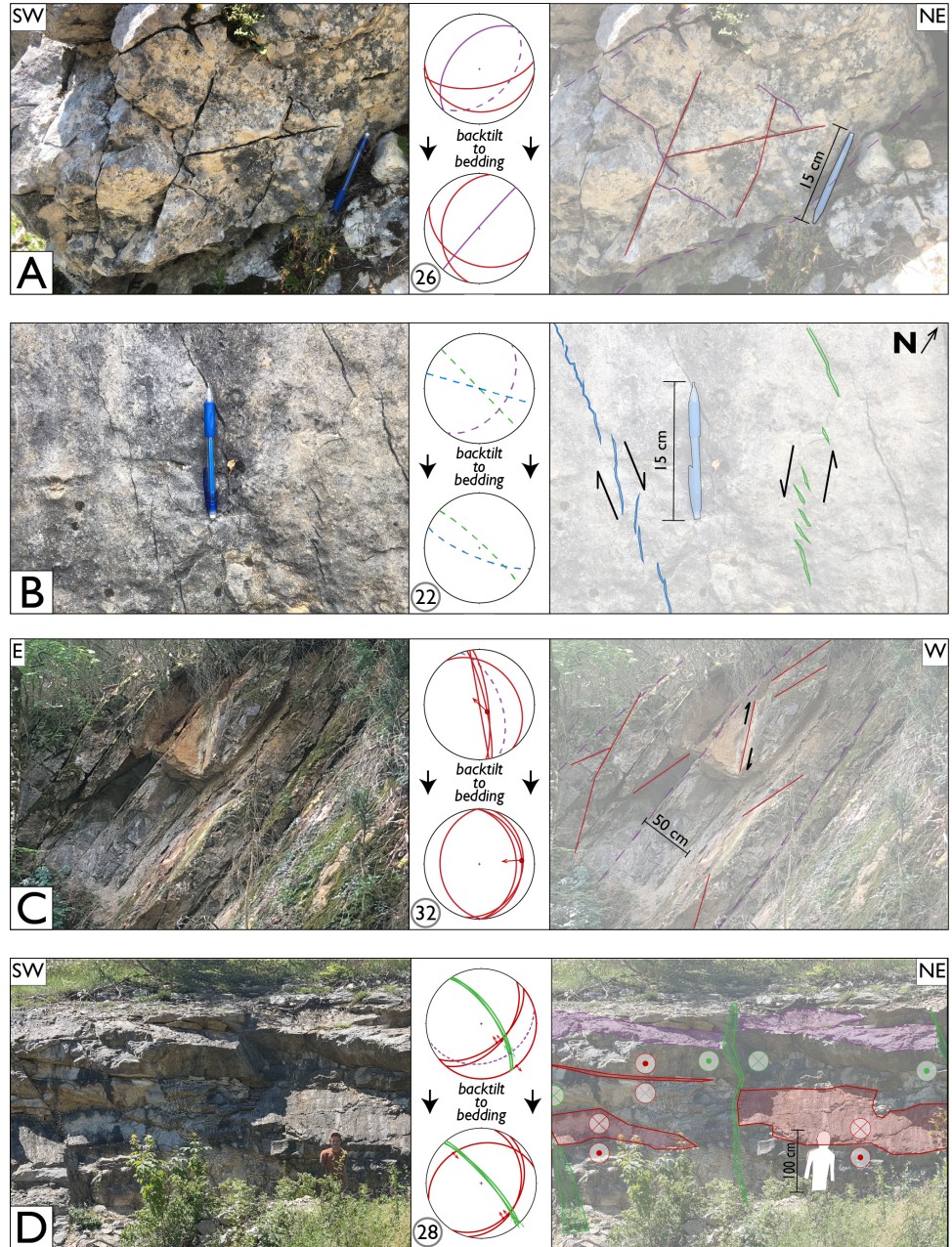

**Figure 6.** Field examples from the Jura. A Reverse small fault with low angle with respect to the bedding, in association with a bed-perpendicular stylolite together form the reverse association JA1. B Top view of a bedding surface displaying a conjugate pair of vein arrays. Together with similarly oriented bed-perpendicular faults they form a strike-slip association JA2. C Tilted small-scale faults at low angle with respect to the bedding. The strike after backtilting is ∼N-S. They form the reverse association JA3. D Reverse fractures of JA1 are cross-cut by bed-perpendicular, sinistral faults of JA4. The plunge of the slickensides on the shear planes indicates they formed prior to tilting of the bedding. For legend see figure 3E.

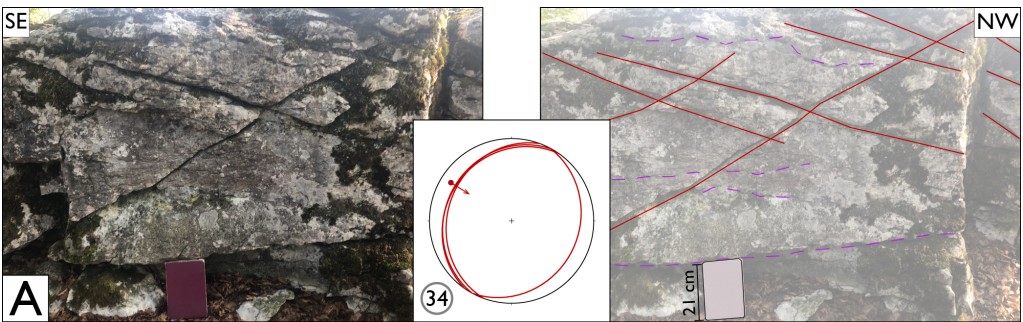

**Figure 7.** Field example from the Salève. A conjugate pair of reverse small-scale faults at low angle with the bedding define the reverse association SA1. For legend see figure 3E.

**Table 1.** Results of the augmented circular scanlines measured on the Parmelan. See figure 4 for the locations of the pavements. Results of circular scanlines of pavement 2 are illustrated in figure 5.

| Pavement | No. of Scanlines (1m radius) | Total discontinuities | PA1 + PA2 | PA3 | Percentage of total |
|----------|------------------------------|----------------------|-----------|-----|---------------------|
| 1 | 9  | 276 | 4  | 79  | 30.1% |
| 2 | 11 | 313 | 39 | 191 | 73.5% |
| 3 | 12 | 207 | 32 | 110 | 68.6% |
| 4 | 6  | 85  | 8  | 39  | 55.3% |
| 5 | 9  | 308 | 31 | 84  | 37.3% |
| 6 | 4  | 150 | 5  | 79  | 56.0% |
| 7 | 4  | 126 | 17 | 70  | 69.0% |

## 5 Improving BHI interpretation

### 5.1 Geothermal exploration wells in the Geneva Basin

We use the prediction of the background network in the subsurface of the Geneva Basin based on the analogue outcrops for the interpretation of BHI of two geothermal wells drilled in the basin (GEo-01 and GEo-02, for location see figure 1). The discontinuities of which the orientation fits within the predicted background network are identified in the dataset and subsequently considered as part of the background network. Then, we evaluate the proportion of the background network with respect to all discontinuities observed in the well.

The two investigated wells both penetrate the Lower Cretaceous carbonates, and as there are no cores of the wells, all features are interpreted on the BHI only. In GEo-01, the Lower Cretaceous is observed between 411 m and 533 m (MD). In this well, two types of image logs are acquired for fracture analysis: optical borehole imaging (OBI) and acoustic borehole imaging (ABI). GEo-02 is located 7 km south-southeast of GEo-01 and here the Lower Cretaceous is found at a depth between 770 m

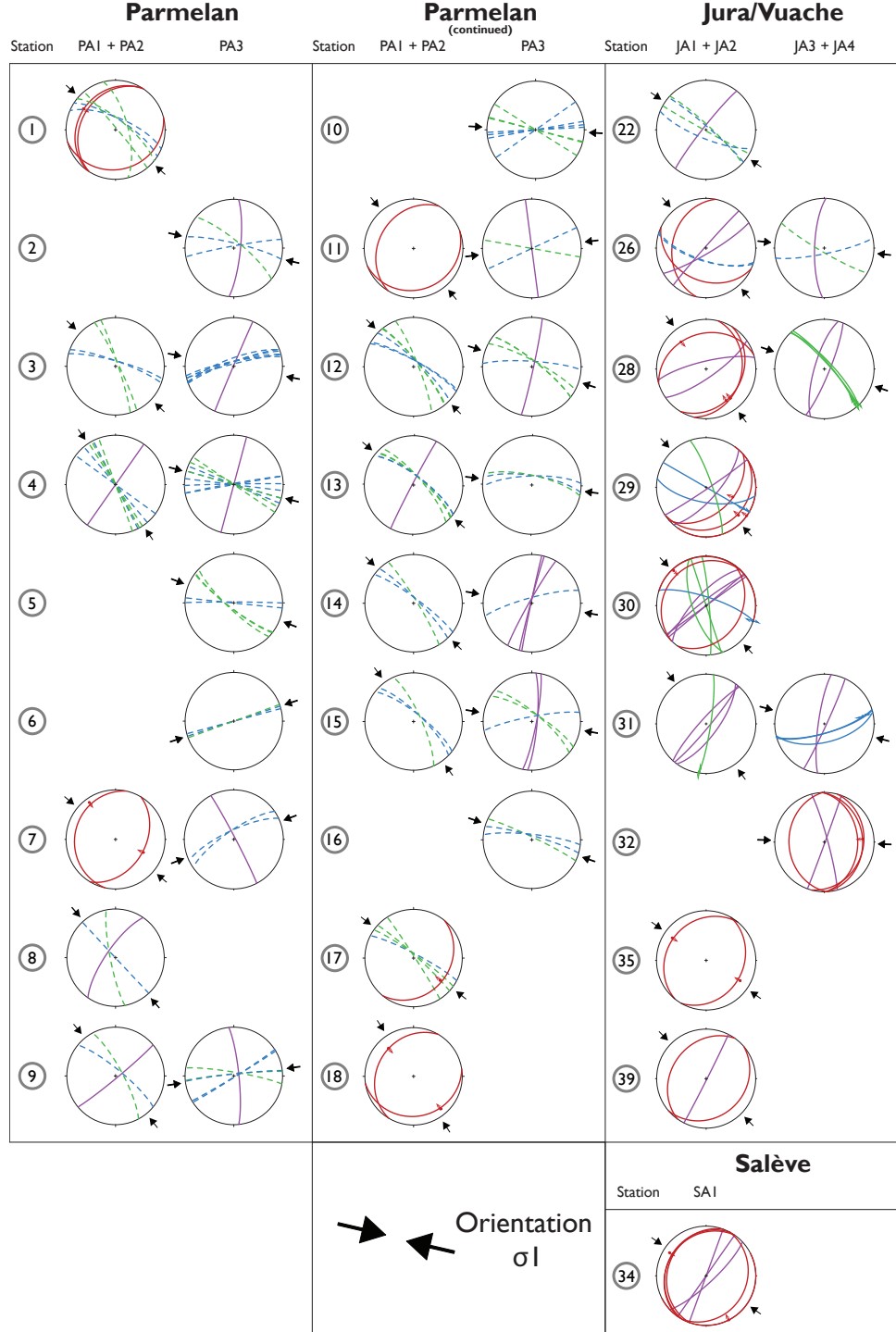

**Figure 8.** Stereonets of the events documented in all the stations of the analogue outcrops. All data is backtilted with respect to the bedding. The black and white arrows are the inferred orientations of $\sigma 1$ of E1 and E2 respectively, and correspond to the arrows plotted on figure 10. For the legend of the colours, see figure 3

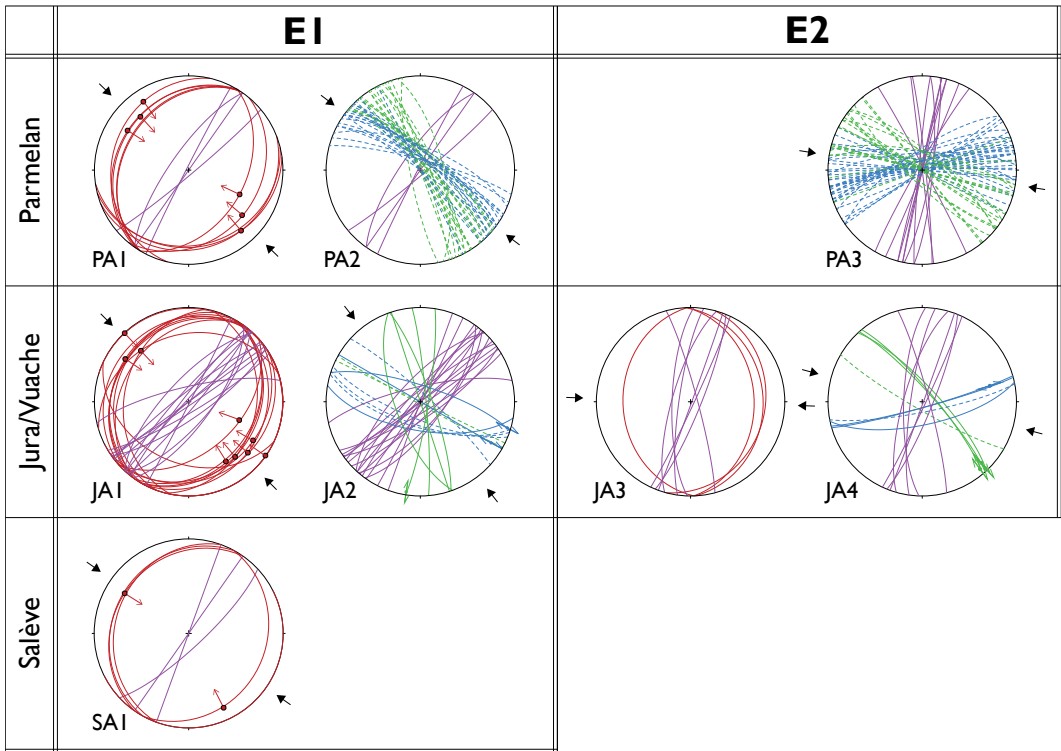

**Figure 9.** The associations of the different outcrops grouped into regional pre-tilting events, based on the orientation of $\sigma 1$. In all outcrops, a reverse association with $\sigma 1$ oriented ~NW-SE is observed (PA1, JA1, SA1). In the Jura and Parmelan, a strike-slip association (PA2 and JA2) with similarly oriented $\sigma 1$ is documented, and together they are grouped into E1. E2 is predominantly a strike-slip association, complemented by a reverse association in the Jura/Vuache. E2 is not observed in the Salève, presumably due to a lack of exposures. All data is backtilted with respect to the bedding. For legend see figure 3E.

to 996 m (MD). For this well, only ABI-logs are available. Feature picking on these logs was carried out with WellCAD (ALT) program software.

The feature picks are divided over 5 different categories; bedding, veins, open fractures, induced fractures and unclassified
fractures. Bedding planes are defined by their repetitive character and the fact that they cannot cross-cut any other feature. Veins are highly reflective in OBI, whereas open fractures are transmissive. Veins were not observed on the ABI logs, either because they are not present, or due to the limited contrast between the host rock and fracture infill. As GEo-02 only has an ABI-log, no veins are interpreted in this borehole. If features are transmissive in OBI, have an irregular surface and the dip angle is high (>85 degrees) they are classified as induced fractures, although it should be noted that separating induced fractures from
natural fractures remains a challenge (Lorenz and Cooper, 2017). Features that are both transmissive in OBI and have a low amplitude on the ABI are interpreted as open fractures. Distinguishing mode-I from mode-II is not possible on the image logs, so they are not differentiated during the picking of fractures. If a fracture pick did not meet any of the above criteria, it was not classified.

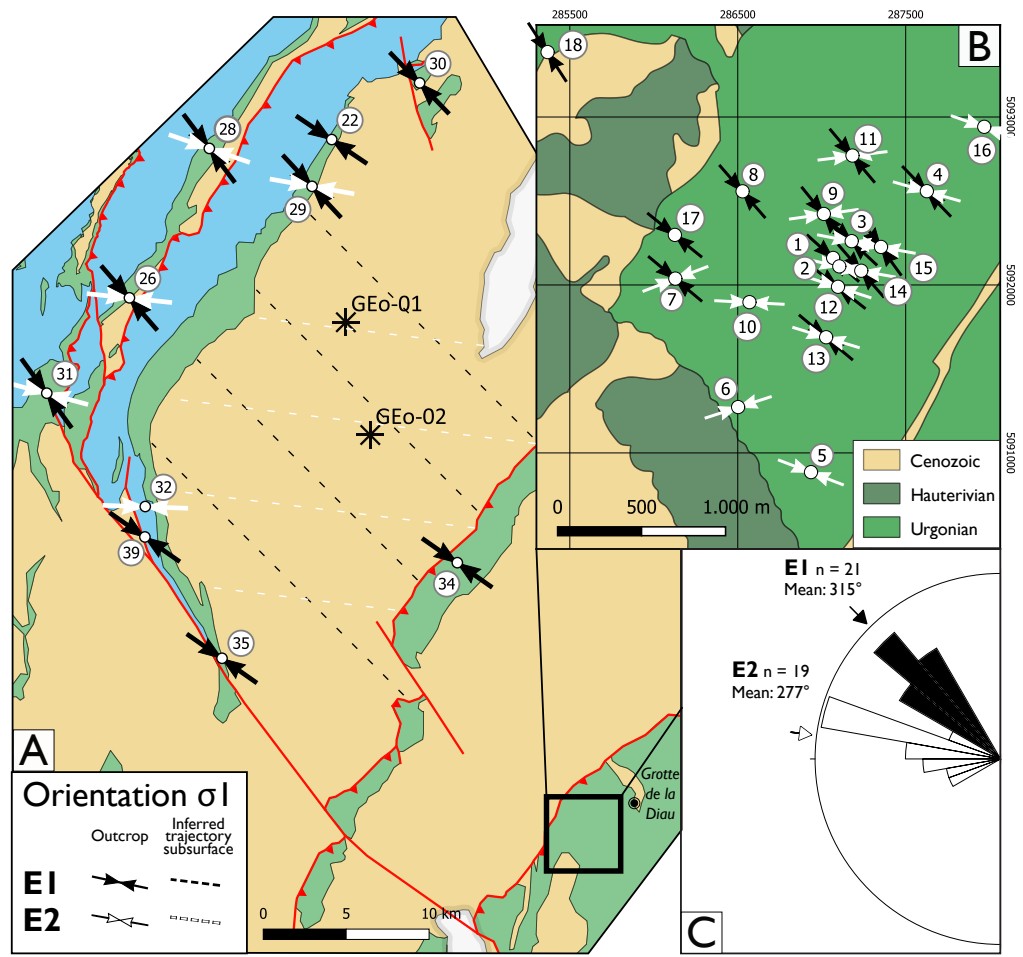

**Figure 10.** Overview of the mapped orientation of $\sigma 1$ of the regionally defined events E1 and E2 recorded in A) the Jura, Vuache, Salève and B) Parmelan C) Rose diagram of the orientations of $\sigma 1$ showing that there is a $\sim 45°$ anticlockwise rotation between E1 and E2.

Firstly, the bedding planes are separated from the other interpreted features. In GEo-01, a total of 195 bedding surfaces are identified (figure 11). When plotted against depth, the dip direction of these planes exhibit a clockwise rotation with depth, from north-dipping at the top, gradually transitioning into south-dipping at the bottom. Also, the dip angle varies with depth, with a low angle at the top and bottom, but a gradual increase between 440-480 m (MD) up to 60°, giving an overall bell-shaped curve. In GEo-02, a total of 176 bedding planes are picked (figure 11). In contrast to GEo-01, the dip direction remains constant with a dip towards the ESE and a dip angle between 10-20°.

For the implementation of the prediction based on outcrop work, we only considered the natural fractures (open fractures and veins), and discarded the induced and unclassified picks from the datasets. Then, the total amount of natural fractures observed in GEo-01 is 820, and for GEo-02 is 211. To compare these discontinuities with the outcrop prediction, they were backtilted

with respect to the closest bedding measurement on the BHI (figure 12). The backtilted discontinuities are subsequently compared with the predicted background network as defined in the outcrops.

## 5.2 Comparison with DAs observed in the field

The orientations of the regional events in the outcrop are used to predict the geometry of individual discontinuity sets that make the background network in the subsurface. We consider the average orientation of the principal stresses of the regional events, and define orientations of discontinuity sets related to this orientation. For E1, the orientation of $\sigma 1$ is 135-315° and for E2 it is 097-277° (figure10). For the low angle small-scale faults, we assumed a strike perpendicular to $\sigma 1$ with a dip angle of 30 degrees. For the stylolites, the strike is also perpendicular to $\sigma 1$, but with a dip angle of 90 degrees (perpendicular to the bedding). As in the outcrop, the majority of the discontinuities that make up the strike-slip associations are bed-perpendicular vein arrays, we considered a 15 degree angle between $\sigma 1$ and the strike of the vein arrays for the prediction in the well. Lastly, the opening fractures are bed-perpendicular with a strike parallel to $\sigma 1$. The combination of these sets are used to identify the fractures in the well that fit within this predicted assoiations. If a fracture pick on the BHI deviates less than 30° (azimuth + dip) from the predicted orientation of a set, these fractures are considered as part of that discontinuity set (figure 12). This maximum deviation is considered reasonable, as there is some variability in the observed DAs in the outcrop. On top of that, there is a margin of error in the BHI interpretation as well.

For the well GEo-01, 350 discontinuities (44% of total) fit in the associations of E1 and E2, of which 236 are open fractures, and 114 veins. In GEo-02, the total number of discontinuities that fit the predicted associations is 104 (50% of total). As there is only ABI for Geo-02, all the fractures that are identified are open. The contribution of E1 and E2 is about equal. The majority of the discontinuities in the two wells are low-angle small-scale faults (175 and 86 for Geo-01 and Geo-02 respectively).

## 6 Discussion

### 6.1 DA-method as analogue link

The regional events captured by the method of DAs reveal similar paleostress orientations as previous studies focusing on the deformation history of the Parmelan (Berio et al., 2021) and the Jura and Vuache ranges (Homberg et al., 1999, 2002). In the Parmelan, Berio et al. (2021) define two pre-folding events with a reverse and strike-slip component with an ∼NW-SE orientation of $\sigma 1$, similar to E1. These authors also document a strike-slip event with an ∼E-W oriented $\sigma 1$ identical to E2 but interpret this event to be post-folding (Berio et al., 2021). In the Jura and Vuache ranges, the reverse and strike-slip regimes of E1 are also observed by Homberg et al. (1999). However, the inferred timing of these regimes is different; only the strike-slip component is interpreted as pre-tilting of the strata, whereas the reverse regime is considered syntectonic, because the majority of the observed reverse slip vectors were not pre-folding (Homberg et al., 2002). This is in contrast to our observations of clearly pre-folding small-scale faults related to the reverse regime (fig. 6A).

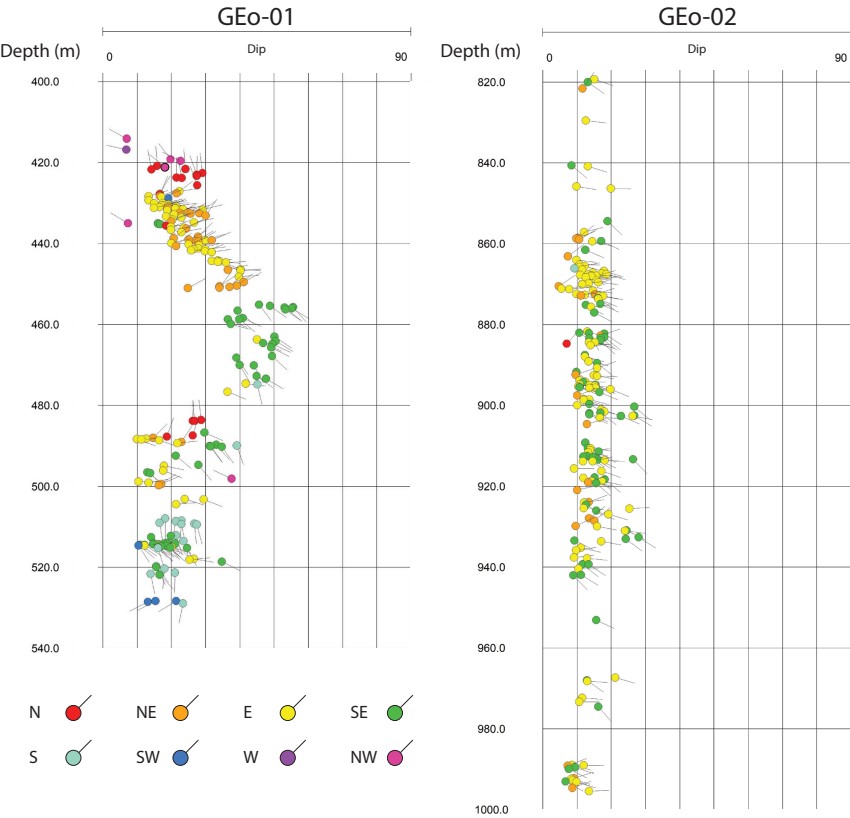

**Figure 11.** Tadpoles of bedding picks versus depth on GEo-01 and GEo-02 after Doesburg (2023). In GEo-01, there is a variation in both azimuth and dip with depth. The bell-shaped curve suggests the presence of a fold, potentially related to a fault around 480 m depth. This contrasts with GEo-02, where no change is observed in bedding orientation with depth.

The main difference between the DA method and previous studies is the interpreted timing of discontinuity-forming events, which reflects the specific aim of the methodology proposed in this study. The goal of the DA method is to use the outcrop

as an analogue of the subsurface to better predict the geometry of the background discontinuity network present at depth that is not directly observable. This is in contrast to the studies of Berio et al. (2021) and Homberg et al. (2002) that focus on the deformation history of the outcrop itself, aiming at retracing the chronological succession of events that created the discontinuities in the outcrop. The consequences of these different approaches are best illustrated with the reverse regime of E1. If this event is linked to folding, it may be expected to produce localized deformations, opposed to the distribution of a

background network (Watkins et al., 2015b). However, this event, producing reverse small-scale faults, could also be interpreted as having formed during the layer-parallel-shortening (LPS) phase before the onset of regional folding. Lacombe et al. (2021) dated the sequence of events shaping a series of folds observed in the Apennines, Pyrenees and in the Rocky Mountains. They found that the LPS-phase largely predates the onset of localized deformations occurring during fold growth and late stage

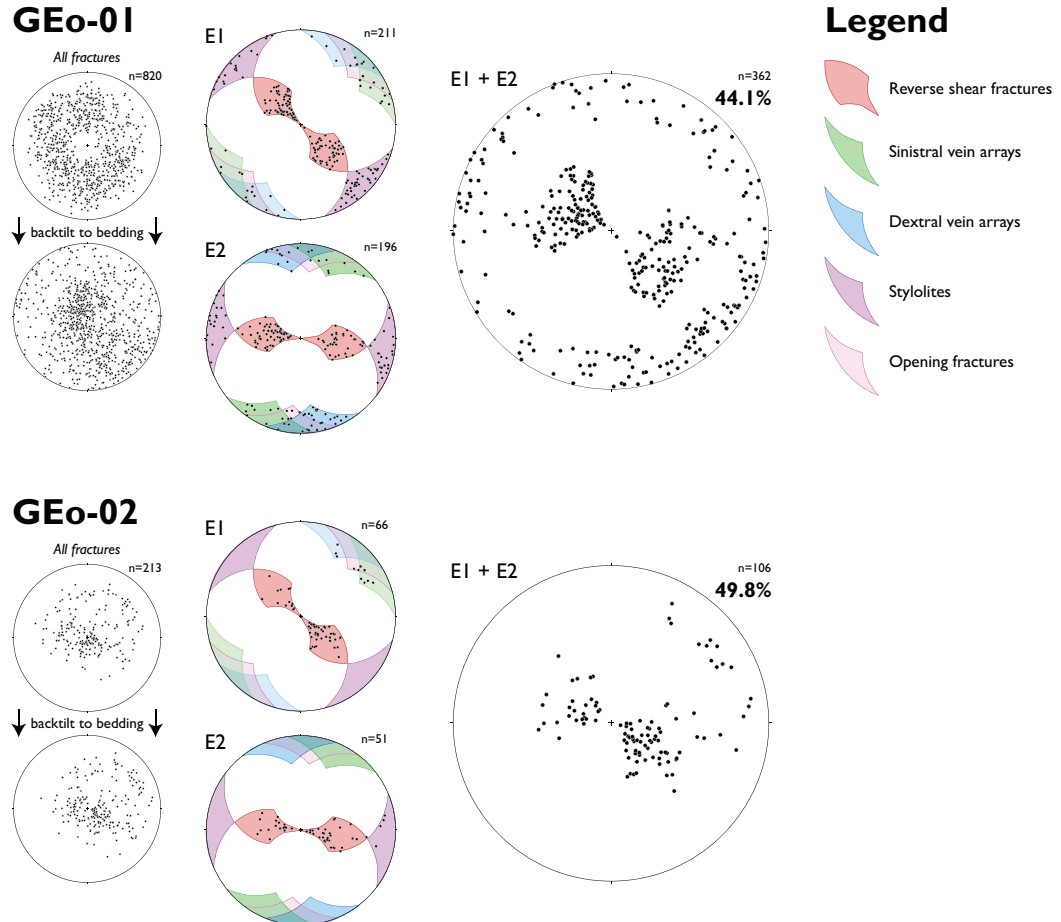

**Figure 12.** All fractures interpreted from the BHI of GEo-01 and GEo-02 by Doesburg (2023) compared with the predicted associations based on the outcrops. First, the fractures in the well are backtilted with respect to the bedding, as the predicted associations are formed pre-tilting. Then, the fractures that fit in the predicted associations of E1 and E2 are counted. The coloured areas are based on the average orientations of $\sigma 1$ of the two regional events (figure 10C), with a deviation margin of 30° (azimuth + dip). There is a small overlap between some sets of E1 and E2: fractures are not counted double when calculating the total percentage of the predicted fractures.

fold tightening. Therefore, the LPS deformation phase is likely to produce the same diffuse fractures observed in the Vuache and the Jura. On the other hand, previous work in the Jura belt, northeast of the area of interest of this study, has shown that stress perturbations impacted the orientation of deformation structures around major left-lateral strike-slip faults (Pontarlier and Morez faults; Homberg et al., 1997, 2004). These faults are similarly oriented as the Vuache fault that bounds the Geneva Basin on the southwestern side. We document two stations in the vicinity of this fault (station 32 and 39, see figure 10), where the orientation of E1 is rotated 10° counterclockwise with respect to the average orientation of E1. It is possible that this small rotation is related to a perturbation of the stress field around the Vuache fault. However, the relationship of E1 with the bedding indicates that it was formed prior to tilting of the strata, and therefore we consider it to be part of the background network. Considering the discontinuities related to E1 as localized and restricted to fold and/or fault structures dramatically changes the prediction of these features in the subsurface. In the absence of folds in the subsurface, these fractures will be overlooked, although they are present in the investigated well in the Geneva basin where no folds or faults are observed (i.e. GEo-02, see fig 12).

A key aspect of the DA-method is that it targets to capture regional events to enhance the predictability for the subsurface. The interpretation of E2 demonstrates this capability of the DA-method. On the Parmelan itself, there is no direct evidence for the relative timing of E2 with respect to the folding. In the Jura however, small-scale faults of E2 are consistently tilted with the bedding (e.g., figure 6C) and thus formed prior to tilting. Therefore, we consider that the simplest interpretation is that E2 is also part of the background network, and consequently predict its presence in the subsurface. Absolute dating of the calcite veins that are part of the background network could potentially constrain the timing even further, but so far, geochronology studies in the region have only focused on dating fault activity (Smeraglia et al., 2021; Looser et al., 2021).

## 6.2 Infill and aperture of discontinuities

The DA-method can be used to predict the geometry of the background network in the target reservoir, but is limited in extrapolating the aperture and mineral infill of fractures. The geometry is useful when considering stimulating the reservoir, as even the sealed discontinuities may create a strength anisotropy that will control the orientation and propagation of hydraulic fractures (Cao and Sharma, 2022; Rysak et al., 2022). However, for predicting flow behaviour in the reservoir caused by natural discontinuities, modeling the aperture and mineral infill of discontinuities is crucial, as only (partially) open discontinuities might contribute to the flow. At the same time, outcrops should be treated with care when extrapolating these properties to the subsurface (e.g. Bauer et al., 2017; Peacock et al., 2022), also when the link between outcrop and subsurface is established with the DA-method. The timing of fracturing, emplacement of the infill and potential dissolution are important factors to consider when extrapolating these characteristics to the subsurface. On the Parmelan, for example, many small-scale (<10 meter) fractures of E1 and E2 are calcite filled (e.g. see figure 3). The diagenetic evolution can be used to constrain the timing of calcite cement formation in the outcrop (e.g. Lavenu and Lamarche, 2018; La Bruna et al., 2020), and subsequently provide insights how the aperture of these discontinuities can be modeled in the subsurface (Elliott et al., 2025). On the other hand, the large-scale fractures (> 100 m) of E1 on the plateau are currently conductive due to dissolution and karstification (see figure 4). It depends on the timing of fracturing and subsequent dissolution if the conductivity of these fractures can be used as an

analogue for the paleokarst network that is observed on top of the Lower Cretaceous in the subsurface of the Geneva Basin (Eruteya et al., 2024). If E1 was formed prior to sub-aerial exposure of the Lower Cretaceous during the Paleogene, it is likely
that they partially controlled the orientation of karst development. On the contrary, if the karstification on the Parmelan only occurred after the exhumation in the Pliocene, similarly dissolved fractures cannot be expected in the subsurface. So, in order to predict the aperture and if discontinuities are sealed in the reservoir, solely based on outcrops, the timing of fracturing and the diagenetic evolution of the formation are both essential to predict which discontinuity sets in the subsurface are likely to be conductive. Another possibility is to use borehole data to assess which discontinuities are conductive, and the DA-method can
be part of the workflow to improve the interpretation.

## 6.3  DAs improve fracture interpretations of BHI

Borehole images are a practical, widely used, and relatively inexpensive way to sample and characterize the sub-seismic scale discontinuity network in the subsurface. However, there are two main drawbacks to this type of data. Firstly, core-to-log correlations have shown that image logs only are not suitable for characterizing the type of fracture (e.g. Laubach et al., 1988;
Genter et al., 1997; Fernández-Ibáñez et al., 2018). Secondly, image log interpretations are prone to subjective bias of the interpreter (Zarian and Dymmock, 2010), similarly as has been demonstrated for fault interpretation in seismic data (Bond et al., 2007) and fracture data collection in outcrops (Andrews et al., 2019; Peacock et al., 2019).

DAs can complement BHI interpretation by providing the discontinuity type of identified background features. Typically, discontinuity sets defined on BHI (in particular when cores are not available) are all considered as opening-mode fractures.
Based on this assumption, a classical workflow consists of defining fracture sets, extracting statistical distributions for these sets, and stochastically extrapolating these distributions at the reservoir scale in a discrete fracture network model (e.g. Hosseinzadeh et al., 2023). However, the type of discontinuity will impact the evaluation of the flow behaviour of the network in multiple ways. Several studies have demonstrated that stylolites can be either flow conductive or form flow barriers and could potentially induce compartmentalization in subsurface reservoirs (Heap et al., 2014; Koehn et al., 2016). Hooker et al. (2012)
and Lander and Laubach (2015) showed that opening fractures are good flow conductors if cement bridges create a natural propping mechanism in the fracture. Finally, the roughness of a discontinuity, which is related to the type, has an impact on its capacity to be reactivated under present-day stress field, which in turn influences its hydraulic aperture under reservoir conditions (Bisdom et al., 2016). These authors also add that typically, shear fractures have a higher roughness than opening fractures, therefore highlighting the importance of being able to constrain fracture type in the reservoir.
The DA-methodology provides a prediction of the discontinuity type of up to 50% of the observed discontinuities in the two boreholes in the Geneva Basin, even though the resolution of the BHI is too low to determine discontinuity type, and there is no core available to correlate the BHI with. The percentage of background-related discontinuities is similar in the two wells, whereas the total number of features is higher in GEo-01. A possible explanation is that there is only ABI-log available for GEo-02, and no OBI. It might be that not all discontinuities present in GEo-02 are visible on the ABI log, due to
a lack of contrast in acoustic properties between host rock and discontinuity, resulting in a lower number of features picked. Another possibility is that the difference in number of background-related features is caused by the spatial variability within

the background network. GEo-01 might have penetrated a denser part of the background network compared to GEo-02. After the emplacement of the background network, GEo-01 is affected by localized deformation (i.e. a fold, see figure 11), producing more discontinuities in this well. At present-day, the percentage of background-related discontinuities is then the same as in GEo-02.

To reduce biases in fracture interpretation in general, Andrews et al. (2019) propose to develop a clear sampling strategy before the actual interpretation. The DA-methodology provides a guideline for such a strategy. For example, in the Geneva Basin, the two events that shape the predicted background network are composed of discontinuity sets with a known range of orientations, based on outcrop work. An adequate fracture picking strategy in the BHI could be to initially discard all features that fall significantly outside this predefined range. This strategy will isolate the background network from more recent discontinuities produced by local drivers. This separation can subsequently be considered during fracture modeling on the reservoir scale. After this, the impact of the background network on the flow behaviour of the reservoir can be assessed with flow simulations.

### 6.4    Impact on fracture modeling for geothermal exploration

The tectonic driver of the background network is fundamentally different from the rest of the network, and therefore isolating the background network in the reservoir will improve fracture modeling on reservoir scale. Maerten et al. (2016) developed a method that links discontinuities observed in the well with seismic-scale faults. A given number of random far-field stress states are simulated around the faults, and the perturbation of the stress directions around the faults is calculated. For each simulated stress state, the number of small-scale discontinuities whose orientation fits within the modeled stress field is counted (goodness of fit). The stress state with the highest number of fitting discontinuities is considered the best stress regime, and the discontinuities falling outside this model are discarded from the dataset. The input data for these models are generally all the fractures interpreted from wells, or, in other words, it is assumed that all subsurface fractures are fault-related. Instead, we propose to first isolate the background network, as these discontinuities should be extrapolated to the entire reservoir. Only after this separation, the goodness of fit of fault-related discontinuities should be considered. In this way, the geological understanding of discontinuity formation is better incorporated in the fracture modeling in the reservoir.

Another way how the DA method can improve fracture modeling in the reservoir is in the up-scaling strategy. Berre et al. (2019) advocated for mixing explicit and implicit representation of fractures in the model as an effective up-scaling method, as it balances the accuracy of the process, whilst preserving the geometrical complexity of the network. Typically, the selection criterion between implicit and explicit representation is the length of the fractures (Lee et al., 2001). As an alternative to this method, we propose to use the genetic origin of the fracture as a second criterion. Due to its regional character, the background network is very suitable for up-scaling strategies. By assessing the impact of the background network on the effective permeability on reservoir scale, either by analyzing the topology of the network (e.g. Sanderson and Nixon, 2015; Hardebol et al., 2015), or by numerically simulating flow through stochastically generated DFNs (e.g. Agbaje et al., 2023; Kamel Targhi et al., 2025), the decision can be made to either represent the background explicitly or implicitly in reservoir scale models. After the significance of the background network is defined, the next step is to include the discontinuities observed in

the well that could not be placed in the framework of the background network. These discontinuities are thus likely created by local drivers and scale differently on the reservoir scale than the background network. For example, if there are seismic-scale faults present in the subsurface, the above mentioned method of Maerten et al. (2016) is a suitable approach to extrapolate these discontinuities to the reservoir scale.

This dynamic workflow will de-risk future geothermal drilling projects in different ways. The separate modeling of the permeability of the background network can be used to assess whether the background only can already produce economically viable fluid volumes, or if seismic-scale discontinuities are essential for production. Also, the well-placing strategy can be adjusted to the heterogeneity of the background permeability field. For example, in the Geneva Basin, most of the background discontinuities are striking NE-SW, and thus, a higher permeability in that direction is expected. A deviation of the well

perpendicular to this strike will therefore likely optimize the well screen and thus the fluid inflow.

## 7   Conclusions

In this study, we presented a novel approach to connect outcrop studies of discontinuities with subsurface characterization of discontinuity networks. Associations of genetically related discontinuities that form the background network produced by the far-field paleostress are defined in the field. The regional character of the background network provides a robust link between

415 analogue outcrops and subsurface target reservoirs. This link is used to improve interpretations of borehole images. By applying this methodology to analogue outcrops of a naturally fractured geothermal reservoir in the Geneva Basin, we have shown that:

1. Discontinuity associations are useful paleostress indicators that enable the reconstruction of the paleo stress field in which the background discontinuity network is formed

2. The regional character of the background network makes it a robust link between the outcrop and the subsurface

3. Analogue outcrops of the Lower Cretaceous carbonates in the Geneva Basin reveal two regional discontinuity-forming events that occurred before Alpine fold-and-thrusting

4. 40-50% of discontinuities observed on BHI from the target reservoir can be explained by the regional events that formed the background network, constrained by the work done on analogue outcrops

5. Outcrop studies may provide a first-order evaluation of the contribution to flow of the background network in the sub-

425 surface.

*Data availability.* The measurements displayed in the stereonets of figure 8 are provided as a supplementary dataset. UAV-derived orthorectified images of the Parmelan (see figure 4) are available via Hupkes et al. (2025).

*Author contributions.* JH: Writing - original draft preparation, Conceptualization, Methodology, Investigation, Data Curation, Visualization; POB: Conceptualization, Supervision, Writing - review and editing; GB: Conceptualization, Supervision, Writing - review and editing; MD: Formal Analysis; AM: Supervision, Writing - review and editing

*Competing interests.* The authors declare that they have no conflict of interest.

*Acknowledgements.* We would like to thank the Service Industriel de Geneve for making the BHI-data of the two wells in the Geneva Basin available to us. The first author is grateful for the Molengraaf Fonds for providing financial support for the conducted fieldwork. The assistance of Nil Feliu during the collection of the circular scanline data is highly appreciated. Didier Rigal and Jean-Marc Verdet are kindly thanked for sharing their endless knowledge on the Parmelan cave system, and for their guidance during the excursion in the Grotte de la Diau. We would like to thank Stephen Laubach and an anonymous reviewer for their constructive comments.

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
