# Peer review of "A Fracture Rarely Comes Alone: Associations of Fractures and Styolites in Analogue Outcrops Improve Borehole Image Interpretations of Fractured Carbonate Geothermal Reservoirs"

_EGUsphere, 2025_

## Author Comment (AC1)

**General comments**

Optimizing the use of outcrops and image log data to characterize subsurface fractures is a topic of widespread scientific and practical interest. This paper provides a useful example of fracture description that is relevant to geothermal applications. The approach is to obtain kinematically meaningful fracture and stylolite relations in various outcrops to identify regionally persistent patterns of fractures. Then the patterns are used to guide image log interpretation in two wells. Using fracture relationships for kinematic interpretation of fracture and stylolite patterns has long precedent (see the review by Hancock, 1985) but the specific approach used here of high grading a suite of key features from several outcrops regionally and the application to comparing outcrop fractures to the subsurface and as an aid to interpreting image logs for geothermal studies is sufficiently innovative and topical to be of interest. The work is within the scope of this journal. The illustrations are good, and the text is mostly well prepared. At 28 pages the MS is succinct.

The MS as it currently stands needs to be improved with moderate revision if the work is to have impact. The Introduction does not make the aims, assumptions, and claims sufficiently clear. Some of the material here can be reorganized. And the claims need to be made explicit. The Methods mix in arguments about interpretation that belong in the Discussion but leave out specifics of what methods were used and contains vague statements about the attributes of the outcrops. The use of well data is not evident from the Introduction or Methods even though two of the MS conclusions focus on claims about the subsurface. A short new but information-rich Methods section is needed. Stylolites are prominent in the MS title and the methods use fracture and stylolite relations, but because of the 'discontinuity' terminology the role of stylolites is not apparent until well into the MS. Although the approach of selecting key fracture relations seems to have worked, I don't think the MS does a sufficient job of discussing the caveats—why did the method work here? What could go wrong if the approach is attempted elsewhere? And the MS needs to do a better job comparing or at least contextualizing the results with other recent attempts to use outcrops to guide fracture assessment for geothermal applications. And some parts of the latter part of the Discussion could be condensed as currently written they do not seem to be well linked to the results.

I appreciate the attempt in the MS title to grab the readers' attention. But the implicit claim of the title: 'A Fracture Never Comes Alone: Associations of Fractures and Stylolites...' is not sustainable. In the literature there are many fractures described that are not associated with stylolites. I don't think that the 1988 fracture review by Pollard and Aydin even mentions stylolites. Plenty of fractures 'come alone'. In many cases where stylolites are present, they are bed-parallel structures that are not necessarily associated with fracture

formation and that do not have usable kinematic significance for fracture analysis. The association of fractures are stylolites is not unique to this example, but the pattern is also not universal. The kind of stylolite appealed to here (MS fig. 1), effectively a kind of widely spaced disjunctive cleavage, is a feature of certain carbonate rocks (and rarely, sandstones) and settings, but the MS (and MS title) never makes this much narrower scope clear. The title of the MS ought to be revised to reflect this. For stylolites, in addition to the Hancock review, the authors should consider the classic paper on the topic by Marshak and Engelder (Marshak, S., & Engelder, T. 1985. Development of cleavage in limestones of a fold-thrust belt in eastern New York. *Journal of Structural Geology*, 7(3-4), 345-359.) Posing the issue as one of kinematic analysis rather than in terms of stresses (or paleostresses) may have some advantages (e.g., Friedman, 1964; Groshong, 1988; Marrett and Peacock, 1999). Marrett, R., & Peacock, D. C. (1999). Strain and stress. Journal of Structural Geology, 21(8-9), 1057-1063.

The introduction hase undergone significant reorganization, to make the statements of our manuscript clearer. The Methods is revised to fit the purpose of this study better, and is placed directly in front of the results, to improve readability. A paragraph in the Discussion is added to discuss limitations of the methodology presented, whereas other paragraphs are condensed. Also, we replaced the word 'Never' in the title for 'Rarely'. We will elaborate on all the changes we made in the 'Specific Comments' section below.

The MS's generalization of fracture and stylolite assemblages from figure 1 (from Hancock 1985) also needs to be presented with more nuance. Although all the structures shown in the figure can form together, the literature suggests that in many cases only one or some of the structures are present (see regional studies by Engelder; and the recent geothermal-related outcrop study by Elliott et al.). And even if all the elements are present, they may not be contemporaneous or even related at all. Such an assemblage relationship needs to be demonstrated, not assumed. In other words, opening-mode fractures do not necessarily bisect arrays of small faults or en echelon fracture arrays (or transect stylolites). I think the authors appreciate this, but the point isn't clear from the text.

We agree that the individual discontinuities and subsequently discontinuity sets are defined partly based on kinematics (result of strain). The goal of the current MS is to obtain a 'genetic interpretation of natural structures' similarly to Marrett & Peacock (1999).

In the MS, we follow the proposed methodology of Hancock (1985): 'The orientation of the principal stresses can be determined knowing that at the time of failure an extension fracture is initiated perpendicular to  $\sigma$ '3 and in the principal stress plane containing  $\sigma$ '1 and  $\sigma$ '2, and that conjugate hybrid or shear fractures enclose an acute bisector parallel to  $\sigma$ '1.'

We deal with very small displacements on single discontinuities, and therefore strain axes can be assumed to be similarly oriented as the stress axes. It is true that we don't know the timing of formation of the fractures and stylolites described in the MS. However, if discontinuities fit in the framework described by Hancock (1985), the simplest interpretation is that they formed in the same stress field. In this workflow, the relative timing between individual discontinuities is of subordinate importance. As we know, the duration of the stress field in which discontinuities form is much longer than the time needed for a fracture to grow and stop, relations between single features do not say much. In addition, the question of whether a fracture crosses another one or it shifts to one side or the other, depends on the energy needed to cut the preexisting vein/fracture and the one needed to overcome friction along the preexisting fracture and jump to another place.

**Specific comments**

1. The Introduction could use work to make it more compelling. Currently I don't see a clear statement of claims near the end of the Introduction. I think that the claims are that: two kinematically consistent groups of small faults, opening-mode fractures, and stylolites, called discontinuity associations, can be recognized in outcrops of Lower Cretaceous carbonate rocks around the margin of the Geneva basin. The associations are widespread in outcrops around the basin margin and because the patterns formed in flat lying beds prior to tilting and folding, the associations are likely of regional extent within the basin. Assuming that the discontinuity associations are regionally extensive and present in the subsurface helps differentiate fracture traces on image logs from two boreholes that penetrate the Lower Cretaceous carbonate rocks in the Geneva Basin. Of fractures visible on image logs, ~45% have orientation patterns consistent with membership in the two discontinuity associations. Thus, kinematic analysis of fracture patterns in outcrop—even where outcrops are small—can yield information that can help guide interpretation of sparse subsurface fracture observations.

The Introduction could also do a better job of leading up to the claims. If I have the claims right, these steps should be:

Fracture patterns need to be considered in geothermal applications. This includes
influences on fluid flow but also rock strength in the case of hydraulic stimulation
where even sealed fractures can modify pattern development (e.g. Cao et al., 2022;
Rysak et al. 2022). (The current MS assumes the main role of fractures is to enhance

fluid flow, but from the Results it looks as though most are calcite filled. They may still constitute strength anisotropy. These latter points need to be added to the Discussion).

- Owing to the limitations imposed by sampling the subsurface with wellbores and the small size of fractures (no effective seismic detection) many distributed attributes of fractures in the subsurface can be/have to be inferred from outcrops. The several studies that have discussed how to pick good outcrop analogs (references). What is typically needed or desired from the analog (large, clean exposures where the hard-to-get attributes of fracture length or connectivity can be measured; several references). And a case needs to be made that the fractures in outcrop correspond to fractures in the subsurface. How have others tried to show this? Are outcrops ever an exact match to the subsurface? But what if outcrops are small? And what if for the subsurface you only have image logs, so methods to demonstrate that the outcrop fractures match the subsurface are limited? What can such outcrop provide? Give a lead in to what your study provides.
- Outcrop fracture studies commonly differentiate fracture sets (orientations, mode, relative timing) and kinematically compatible associations of fractures and stylolites (your figure 1) (Hancock, 1985 review). These patterns can commonly be recognized in small outcrops where only part of the pattern is evident. But larger outcrops have the advantage of documenting the patterns more fully. Thus, small outcrops or even subsurface data can be used to identify patterns. Here it would be useful to mention the similar approach of regional studies of coal cleat patterns that were derived from small outcrops around basin margins and subsequently were used to help interpret core, image log, and production patterns in coalbed methane applications in the 1990s.
- From a practical standpoint, the subsurface fractures are what matters. Cost-effective subsurface fracture evaluation commonly relies on image logs. But in addition to the inherent sampling limitations imposed by the small dimension of the wellbore relative to fracture patterns, there is considerable uncertainty in how to interpret image logs (telling natural from drilling induced fractures is a problem that goes back to the inception of these logs in the 1980s; referencing should reflect). Thus, there is a need for approaches that help interpret image logs. How can outcrop studies contribute?
- Here we show that two kinematically consistent groups of small faults, openingmode fractures, and stylolites, called discontinuity associations, can be recognized in outcrops of Lower Cretaceous carbonate rocks around the margin of the Geneva

basin. The associations are widespread in outcrops around the basin margin and because the patterns formed in flat lying beds prior to tilting and folding, the associations are likely of regional extent within the basin. Assuming that the discontinuity associations are regionally extensive and present in the subsurface helps differentiate fracture traces on image logs from two boreholes that penetrate the Lower Cretaceous carbonate rocks in the Geneva Basin. Of fractures visible on image logs, ~45% have orientation patterns consistent with membership in the two discontinuity associations. Thus, kinematic analysis of fracture patterns in outcrop—even where outcrops are small—can yield information that can help guide interpretation of sparse subsurface fracture observations.

With each bullet point equal to a paragraph with background citations.

The introduction has undergone major revision (line16-77). We changed the content and order of paragraphs, following the proposed bullet points, but with some modifications:

- Understanding discontinuity networks is important for geothermal exploitation of fractured reservoirs. The references related to hydraulic stimulation are added (line 21-22).
- Borehole images are an important tool for subsurface characterization of discontinuities but come with limitations. Note that we present this already in the second paragraph to underline that in the end, it are the discontinuities in the subsurface that matter.
- Analogue outcrops can be used to overcome certain limitations of BHI. To be a good analogue, the outcrop must meet several criteria, among which the shared tectonic/stress history.
- To understand the genetic origin of a discontinuity is essential for extrapolation in the subsurface to reservoir scale.
- There are several methods to determine paleostress. One is to group discontinuities into associations, following Hancock (1985), and despite its simplicity, it is barely used to link outcrop with subsurface.
- Statement of claims. ~45% of the features identified on BHI of the target reservoir in the Geneva Basin can be placed in the predicted framework of the background network.

Also, we changed the order of the Geological Background and the Methodology. This to improve readability in the first part of the manuscript.

The focus on needing to find evidence for 'geomechanical drivers' seems like it could be hard to support and is probably not needed. Showing why fractures formed is notoriously

challenging and the evidence for doing so is probably lacking in this instance. Where is the precise fracture timing and depth information or the rock mechanical properties history? For the approach used in this MS to work, that kind of geomechanical argument is probably not needed. The authors use orientation and relative timing information from geometric relations to classify faults, opening-mode fractures, and stylolites into kinematically compatible groupings and show that over a wide area these groupings have consistent patterns. The regionally consistent patterns can then be used to improve interpretation of fractures visible on image logs in some wells.

We agree that 'geomechanical' driver was not the right word to emphasize our point, and replaced it by 'tectonic' driver (see line 40). However, we strongly believe that, to improve subsurface discontinuity modeling from a geological perspective, it is essential to consider the stress fields in which discontinuity associations are formed. Without such a step, we cannot predict if a given discontinuity (association) has to be extrapolated to the entire reservoir (with a spatial variability), or only along a fault/fold. In this study we demonstrate that ~45 % of the discontinuities observed on the BHI should be extrapolated to the entire reservoir, and this with the predicted geometric relations with respect to the bedding. This can only be done when considering the tectonic driver of the discontinuities, and not just the kinematics.

The writing could use work to make the text clearer. I've marked some of these concerns below keyed to lines in the text. The use of terms, particularly in the Introduction, is confusing and the text there and elsewhere ought to be rationalized for clarity. I don't think there is a meaningful difference between 'fractures' and 'fracture sets' and 'discontinuities' and 'discontinuity sets' except that the latter two are less widely used. And, on line 86, the text finally mentions that stylolites are also being measured. The paper title uses 'fracture' and nothing would be lost by sticking with this and related terms throughout, or 'fractures and stylolites'.

We think 'fracture' has been used by different people in different ways, that we want to avoid confusion by introducing a new term for our methodology. We chose the term discontinuity, as we want to include both stylolites and fractures in our proposed methodology. In this study, we do not focus on individual sets, but on the stress fields in which associations of discontinuity are formed. Therefore, we removed the definition of sets from the Introduction and only mention it in the Methodology with references (line 113).

1-8 (Abstract) The text here is all correct but it seems out of place in an Abstract, which I think ought to get to the findings more directly. This text seems more suitable to the Introduction or the start of the Discussion. Consider condensing or moving this text.

The Abstract could start with the text in line 7 (with edits): "We present a method that uses associations of fractures and stylolites, which we call discontinuity sets, to link outcrop and subsurface structures. Discontinuity sets are associations of kinematically compatible structures—faults, opening-mode fractures, and stylolites—that can form broadly contemporaneously. Relative timing can be obtained from crossing and abutting relations. Although such associations are commonly described in outcrop fracture studies they are rarely used to link outcrop observations to structures in geothermal targets or to help guide classification of sparse structural observations made using image logs. We use the orientations and type of discontinuity associations as indicators to map out principal paleostress trajectories of regional discontinuity-forming events that created a background discontinuity network...' (See the comments on the Introduction structure above).

The abstract is adjusted, and the first lines are removed. As mentioned above, we focus not on individual sets, but on discontinuity associations. We prefer not to use 'kinematically compatible', as we think this is somewhat misleading: discontinuities are compatible with respect to the information they deliver on the stress field in which they formed, not for their kinematics. Also, we do not introduce different 'type of discontinuity associations' – we only consider the orientation of associations to determine the stress regime in which they are formed.

8 'robust'? It seems like a plausible link, but what do you mean by robust? Maybe you could claim robustness if you had independent timing information. Seems overstated.

**Accepted. 'Robust' is removed.**

21-22 This line struck me as sounding a bit circular: '[fractures] control...in fractured reservoirs...' It's also potentially confusing and convoluted since most of the 'discontinuities' are fractures but this isn't stated. Can this line be revised to be more straightforward? Also, it might be worthwhile to mention that not all geothermal reservoirs are fractured.

This line is replaced by: 'Carbonate geothermal reservoirs with a low matrix porosity and permeability may still have a convective heat flow due to the presence of natural discontinuity networks (NDNs) (Berre et al., 2019; Medici et al., 2023)', see line 17-18.

22 Berre et al. and Medici et al. both talk about 'fracture networks'. So why call these 'discontinuity networks'? These are also review papers about modeling, where the features

conducting fluid are assumed to be open fractures. That seems to differ from what you are dealing with: partly or fully sealed fractures (veins) and stylolites.

We refer to review papers concerning modelling, as that is the final goal of using outcrops as an analogue of geothermal reservoirs - to apply observations from the outcrop to the subsurface models. It is a good point that we are dealing with (partially) filled discontinuities in the outcrop, and that this impacts the flow assessment in the subsurface - a paragraph on this is added in the Discussion (line 309-331). As mentioned above, we do want to include both fractures and stylolites, hence 'discontinuity'.

24 These La Bruna et al. papers are great (only one of them is in the reference list). But they are about outcrop fractures or fracture attributes that *might* influence flow rather than, as the sentence implies, being about studies that demonstrate with evidence such as production data that fractures actually influence flow. Many papers describe features that might influence flow, so it's not clear why these two papers would be singled out (an e.g., at least is needed). But what you want to support the statement is one of the papers that uses well data to make this point. The number of papers that demonstrate that some aspect of subsurface fractures influence flow is pretty small, owing to the problems of characterizing subsurface fracture arrays. A paper that uses production data evidence wrt fractures is Solano et al. 2011 SPE Res Eval Eng. I suggest that the line and referencing be modified to reflect this. One example in the literature that links a specific fracture attribute to flow response is open versus sealed fractures (e.g. Weisenberger et al. 2019 Petroleum Geoscience). So if your fractures are fully or partly sealed this ought to be addressed in the Discussion.

References are added in line 20 to support this paragraph (Caine et al. 1995, Solano et al. 2011, Grare et al. 2018, Fadel et al. 2023). Sealed, or partially sealed discontinuities will indeed impact the flow in the subsurface. The presented DA-methodology does not directly allow to extrapolate infill and/or apertures of discontinuities in the subsurface. We added a paragraph in the Discussion (line 309-331) to discuss this limitation of the method, and possible ways for future work how to address this.

28 For definitions of fracture sets see the review by Hancock 1985, J. Struct. Geol. Another aspect of sets is 'relative timing'. Why omit it here?

The definition of sets has been moved to the Methodology section, with references (line 112-113). As mentioned above, we prefer not to use relative timing as a criterion for a set, as this does not make much sense in the context of DAs, where multiple sets may form in a single stress field.

29 I suggest that you tone down the geomechanical aspect here as unneeded. All you need to know, or assume, is that the structures are kinematically compatible and broadly contemporaneous. That's what figure 1 shows. You use the relative timing between structures, from crossing and abutting relations, and their orientation with respect to tilted beds to group structures. The claim here is that fractures ought to be separated by 'geomechanical driver' and although this approach has precedent going back at least to Nelson's 1985 book using a mechanism or 'driver' is a problematic way to classify fractures since the cause of fractures is notoriously hard to specify. Fold- and fault-related fractures and regional fractures have been recognized in the literature since at least the 1950s (as call outs to the literature ought to reflect) but unless you already know what the distribution and timing of fractures is how does an appeal to a 'geomechanical driver' help? If you are looking at a fracture in core (or a trace on an image log) you probably will not be able to accurately classify the fracture as 'fold related' or 'regional'. See the discussion of equifinality in Revs. Geophys. 2019. Maybe the driver material belongs in the Discussion.

The challenge is indeed that with borehole images alone, the distinction of the driver cannot be made. That is where we use the outcrops as an analogue. Not every reservoir has suitable analogue outcrops, but for the Geneva Basin specifically they are present. The results of the outcrop study (i.e. the discontinuity associations that make up the background network) are used to make such a distinction. But to do this, it is crucial to interpret the genetic origin, or, tectonic driver of the discontinuity associations, otherwise we cannot extrapolate them to the subsurface.

The entire paragraph from 27 to 34 seems out of place.

Background or regional fractures are not necessarily more evenly or uniformly distributed than other types of fractures. The literature has excellent examples of clustered fractures within regional sets. See the 2018 J. Struct. Geol. theme issue on spatial arrangement for examples.

Accepted. The definition of sets is removed from the Introduction and only mentioned in the Methodology. We added references for the spatial variability of the background network (see line 45-48).

30 'regional' fractures have been recognized in the literature at least as far back as Balk, 1936. Balk, R. (1936). Structure elements of domes. AAPG Bulletin, 20(1), 51-67. And there are studies that identify regional fracture patterns in outcrop and compare them to sparse core observations.

References are added, but we consider recent studies more insightful (see line 40-43).

31 'the' background set. This seems to imply that there might just be one regional set. But regional studies (like papers be Engelder from outcrops in NY) document multiple regional sets.

We refer to the background network. This network can of course exist out of multiple discontinuity sets.

40 The fracture sampling issue needs to be mentioned. Part of this concerns gaps in fracture observations that are inevitable when using wellbores to sample dispersed features like fractures and another is the problem of putting the sparse fracture samples into broader context: in other words, how easy is it, for example, to specify that a trace on an image log corresponds to certain features seen in outcrop? Part of this latter issue is how similar fractures look that formed by different processes (a situation called 'equifinality' where these issues are extensively discussed in a recent review: Laubach et al., 2019, Reviews of Geophysics). Since this MS proposes a solution to this issue by isolating specific kinds of kinematically meaningful relationships from the outcrop and using those geometric and relative timing inferences to guide image log interpretation, it would strengthen the argument to describe this sampling issue explicitly.

**Accepted. We added the point of 'equifinality' and the reference in line 30-32.**

46-53 This paragraph struck me as vague and having a mixed message. The previous paragraph established that wellbore data has limitations. If you have fracture/stylolite relations in core or visible on image logs, that tells you something about the structures in the subsurface that would not obviously be improved by seeing that relationship in a distant outcrop. A useful thing about outcrops is being able to see features that can never be directly observed in the subsurface, like length or connectivity, which by their nature cannot be captured by wellbore probes.

And the referencing could be more extensive. There have been several studies that specifically address the issue of how to compare outcrop fractures to the subsurface, including specifically for geothermal applications. Note them. Or cover the topic, with references, in the Discussion.

What do you mean by 'analogy' and there is more involved in a useful comparison that just similar rock types, age, and structural setting (including diagenesis/rock property history).

For one thing, outcrops by definition have different loading histories than rocks that are still in the subsurface. It's well established that uplift and unloading commonly do produce fractures (e.g., Engelder, 1985; English, 2012) as do a wide range of near subsurface and geomorphic processes (e.g., Eppes et al., 2024, Earth Surface Dynamics 12, 35-66. https://doi.org/10.5194/esurf-12-35-2024) so these differences may not be trivial. The first

step in outcrop fracture studies aimed at guidance for the subsurface is usually trying to identify these.

I suggest you provide a broader assessment of how exposed rocks are judged to be appropriate analogs for the subsurface target (see papers by Agosta et al., 2010; Sanderson, 2016; Ukar et al., 2019). Possibly in the Discussion. A range of factors go into selecting a good analog for a subsurface geothermal target, including matching rock types and—broadly—structural history (Bauer et al., 2017; Busch et al., 2022, Peacock et al., 2022; Elliott et al., 2024). Some studies have questioned the viability of using outcrops for making *specific* predictions about key subsurface parameters I (Peacock et al., 2022) whereas others claim that such assessments are possible in some instances (Elliott et al., 2024). Since what you are doing is a contribution to solving this problem, the Discussion is a good place to contextualize your work. Many of the other approaches such as using chemical aspects of the fracture system (e.g. Elliott et al. 2025) seem like they would be a good compliment to your approach.

- Agosta, F., Alessandroni, M., Antonellini, M., Tondi, E., and Giorgioni, M. (2010).
   From fractures to flow: a field-based quantitative analysis of an outcropping carbonate reservoir. Tectonophysics 490 (3-4), 197–213.
   doi:10.1016/j.tecto.2010.05.005
- Sanderson, D. J. (2016). "Field-based structural studies as analogues to sub-surface reservoirs," in The value of outcrop studies in reducing subsurface uncertainty and risk in hydrocarbon exploration and production, Geol. Soc. Editors M. B. J. Bowman, H. R. Smyth, T. R. Good, S. R. Passey, J. P. P. Hirst, and C. J. Jordan (London: Special Publications) 436, 207–217. doi:10.1144/sp436.5
- Ukar, E., Laubach, S. E., and Hooker, J. N. (2019). Outcrops as guides to subsurface natural fractures: Example from the Nikanassin Formation tight-gas sandstone, Grande Cache, Alberta foothills, Canada. Mar. Petroleum Geol. 103, 255–275. doi:10.1016/j.marpetgeo.2019.01.039
- Bauer, J. F., Krumbholz, M., Meier, S., and Tanner, D. C. (2017). Predictability of properties of a fractured geothermal reservoir: the opportunities and limitations of an outcrop analogue study. Geotherm. Energy 5 (1), 24–27. doi:10.1186/s40517-017-0081-0
- Elliott, S.J., Forstner, S.R., Wang, Q., Corrêa, R., Shakiba, M., Fulcher, S.A., Hebel, N.J., Lee, B.T., Tirmizi, S.T., Hooker, J.N., Fall, A., Olson, J.E., Laubach, S.E. (2025).
   Diagenesis is key to unlocking outcrop fracture data suitable for quantitative extrapolation to geothermal targets. Frontiers in Earth Science 13, 1545052.

Peacock, D. C. P., Sanderson, D. J., and Leiss, B. (2022). Use of analogue exposures
of fractured rock for Enhanced Geothermal Systems. Geosciences 12 (9), 318.
doi:10.3390/geosciences12090318

This paragraph is rewritten and split into two paragraphs (line 33-50). We consider that the 'goodness' of an outcrop as analogue depends on what you use the outcrop for. In our case, we demonstrate that the orientation of the background network- defined as formed prior to tilting of the strata - is very consistent on a regional scale (on all sides of the basin) and formed prior to tilting of the strata. Therefore, these outcrops are a good analogue for the expected background network in the subsurface. Discontinuity attributes which should indeed be derived from outcrop analogue such as length and connectivity are out of the scope of this study, as they would require a different approach to justify if the outcrop might be a good analogue in this case.

With the restructured paragraphs in the introduction, we clarify this by emphasizing the importance of interpreting the genetic origin of a discontinuity before extrapolating it to the subsurface.

We have added additional references:

Agosta et al. 2010, Sanderson 2016, Ukar et al. 2019 (line 34-35).

Bauer et al. 2017, Peacock et al. 2022 (line 35-36)

Elliott et al. 2025 (line 38)

Engelder 1985, English 2012 (line 39)

55 Genetic relations between fractures and stylolites have long been appreciated. See references in Groshong (1975). And that multiple fracture orientations can form in a single deformation goes back at least to Stearns. See also: Olson, J. E., 2007, Fracture aperture, length and pattern geometry development under biaxial loading: a numerical study with applications to natural, cross-jointed systems. In Couples, G & Lewis, H., eds., Fracture-Like Damage and Localization, Geological Society of London, Special Publication. 289, 123-142.

Groshong Jr, R. H. (1975). Strain, fractures, and pressure solution in natural single-layer folds. Geological Society of America Bulletin, 86(10), 1363-1376.

Accepted. Reference is added in line 56

68 'carbonate rocks'; just saying carbonates sounds slangy.

Accepted

80 (Methods section) This section is confusing. The second part of it (2.2) seems like it belongs in the Discussion. In 2.2 you are making the case that your outcrop data can be linked to the subsurface. This is an interpretation, not a method. The point is best addressed in the Discussion.

We do think this part belongs to the Methodology section. It is essential to this study that we do not aim to measure all discontinuities in the outcrop, but only those that can be placed in discontinuity associations. How we link the outcrop with the subsurface, as described in this paragraph, is the justification for this. We think we need to introduce this in the Methodology section, otherwise the way we present the Results makes no sense.

The first part of the Methods (2.1) also needs to be clarified. Mixed in here are incomplete descriptions of the outcrop sizes and what can be measured in them, data collection methods like circular scanlines that may have only been collected at one outcrop (line 100), a distribution of 10×10 m outcrop stations that is supposed to be "...as evenly as possible over the studied area", and a method for selectively extracting kinematically significant fracture/stylolite relations. These elements need to be separated out and described clearly and quantitatively. Some aspects, like outcrop sizes, maybe ought to be in the Geological Setting. A useful approach would be to build a table and use that as a guide to revising the section. The Methods section also should mention that you had access to and described two wells (line 220).

My suggestions above for the Introduction are based in part on the impression I had from this section that the outcrops you had to work with are small, and not amenable to the type of analysis of large clean outcrops as for example in Elliott et al. 2025.

Dimensions of the outcrops are added in the Geological Background (line 86, 90, 92 and 96). Also, the Geological Background is placed before Methodology, so the outcrop dimensions are mentioned before the Methodology. This rearrangement improves the readability of the manuscript. Also, the wells are mentioned in the Introduction and Methodology (line 63-65 and line147-149 respectively).

The different mountain ranges in this study have different outcrop sizes, as now stated in the Geological Background. The Parmelan has the largest surface exposure in the form of a large pavement (~2x2.5 km), but the other regions have smaller outcrops (

The reference is added in the discussion (line 321).

131 'excellent exposures' is vague. How big, how complete is the exposure? Are fractures that formed in the subsurface readily separated from surface-related fractures here? How?

Dimensions of the exposure is added in the Geological Setting (line 86). The DA-method discriminates between surface related and subsurface related discontinuity by using the geometrical relationship of the DA with respect to the bedding. Aperture, mainly by dissolution, might have changed significantly during exhumation of the outcrop to subaerial conditions, and therefore we do not extrapolate this characteristic to the subsurface. A paragraph is added on this in the Discussion (line 310-331).

140-146 (In the Geological Setting) It would be useful to mention, even if qualitatively, how the structural and burial history or outcrops and rocks in the subsurface differ. Also mention the current state of stress/ stress regime (could cite world stress map papers). In some areas surface fractures relate to current stresses (see the pop ups described by Engelder in the 1980s; references in Elliott et al. 2025).

Accepted. The burial history for outcrop and target reservoir are added in the Geological Setting, as well as the current stress state (line 97-108).

149 (In the Results) It might be helpful to start by describing the structural elements that are present in the entire area.

This is done in the Geological Setting. As we present no new results with respect to these major structures, we think that an additional description in the Results section is not needed.

156 Note and consider the strong condemnation of the term 'shear fracture' in the Pollard and Aydin 1988 GSA Bulletin review. Maybe 'small displacement faults'?

We don't think this terminology will clarify the text. 'shear fracture' is widely used and is included in the glossary of Peacock et al. (2016).

168 Consider adding a star or other mark to the stratigraphic column to show which unit is being analyzed.

**Accepted. A star is added in the figure.**

170 Dissolution along the fractures. Does this play a part in the interpretation? This may be of interest to readers concerned with some of the deep carbonate fractured reservoirs in China, where this kind of dissolution is a key element. Is there any evidence of this process in outcrop? This seems like it could be part of your Discussion.

This is a very interesting point, and we added a paragraph to the Discussion. The Lower Cretaceous has been exposed to sub-aerial conditions in the Paleogene, and this has resulted in a karst system, observed on seismic data in the subsurface of the Geneva Basin (Eruteya et al. 2024). However, it remains a question how representative the karsts in the Parmelan outcrop are for the subsurface, as they also have formed during the most recent exhumation. This question is similar as the topic of extrapolationg aperture and/or infill of discontinuities to the subsurface, so added a paragraph on this in the Discussion (line 310-331):

**'6.2 Infill and aperture of discontinuities**

The DA-method can be used to predict the geometry of the background network in the target reservoir, but is limited in extrapolating the aperture and mineral infill of fractures. The geometry is useful when considering stimulating the reservoir, as even the sealed discontinuities may create a strength anisotropy that will control the orientation and propagation of hydraulic fractures (Cao and Sharma, 2022; Rysak et al., 2022). However, for predicting flow behaviour in the reservoir caused by natural discontinuities, modeling the aperture and mineral infill of discontinuities is crucial, as only (partially) open discontinuities might contribute to the flow. At the same time, outcrops should be treated with care when extrapolating these properties to the subsurface (e.g. Bauer et al., 2017; Peacock et al., 2022), also when the link between outcrop and subsurface is established with the DA-method. The timing of fracturing, emplacement of the infill and potential dissolution are important factors to consider when extrapolating these characteristics to the subsurface. On the Parmelan, for example, many small-scale (<10 meter) fractures of E1 and E2 are calcite filled (e.g. see figure 3). The diagenetic evolution can be used to constrain the timing of calcite cement formation in the outcrop (e.g. Lavenu and Lamarche, 2018; La Bruna et al., 2020), and subsequently provide insights how the aperture of these discontinuities can be modeled in the subsurface (Elliott et al., 2025). On the other hand, the large-scale fractures (> 100 m) of E1 on the plateau are currently conductive due to dissolution and karstification (see figure 4). It depends on the timing of fracturing and

subsequent dissolution if the conductivity of these fractures can be used as an analogue for the paleokarst network that is observed on top of the Lower Cretaceous in the subsurface of the Geneva Basin (Eruteya et al., 2024). If E1 was formed prior to sub-aerial exposure of the Lower Cretaceous during the Paleogene, it is likely that they partially controlled the orientation of karst development. On the contrary, if the karstification on the Parmelan only occurred after the exhumation in the Pliocene, similarly dissolved fractures cannot be expected in the subsurface. So, in order to predict the aperture and if discontinuities are sealed in the reservoir, solely based on outcrops, the timing of fracturing and the diagenetic evolution of the formation are both essential to predict which discontinuity sets in the subsurface are likely to be conductive. Another possibility is to use borehole data to assess which discontinuities are conductive, and the DA-method can be part of the workflow to improve the interpretation.'

198 'is composed of' but 'comprises'. You use this weird English convention correctly in line 190.

**Accepted.**

224 (figure 6) Nice way to do the scales on these images.

**Thank you kindly!**

225 These wells need to be anticipated in the Introduction and Methods.

Accepted. The title of the paragraph is changed into 'Geothermal exploration wells in the Geneva Basin'. Also, the wells are now presented in the Introduction (line 63-65) and Methodology (line 148-149).

232 Help the reader understand the Doesberg 2023 reference (an unpublished MS thesis). Did you do image log interpretation or just use some kind of compilation from this reference? Line 236 makes it seem like you interpreted the images. You might be interested in how Wang et al. 2023 handled references to reinterpreted archival image log data: Wang, Q., Narr, W., Laubach, S.E., 2023. Quantitative characterization of fracture spatial arrangement and intensity in a reservoir anticline using horizontal wellbore image logs and an outcrop analog. Marine & Petroleum Geology 152, 106238.

https://doi.org/10.1016/j.marpetgeo.2023.106238

Indeed, a good point. We removed the reference, as M. Doesburg is a co-author of the manuscript. Doesburg was a MSc-student that carried out the picking of the image logs within the context of her MSc-thesis, under supervision of P-O. Bruna, G. Bertotti, A.

Moscariello and J. Hupkes. We consider her contribution as such that she is a co-author of the current MS, instead of referring to the unpublished MS thesis.

239 How can you know that veins are 'invisible' if you don't have core? Filled fractures do commonly show up on image logs.

That is true. Probably, the contrast between the infill and host rock is too small, but we cannot exclude the possibility that they are not present at all. We added this in line 240-241.

239 'feat' > feature?

Corrected (line 242).

250-252 Hmm. What if these picks are wrong? Is this discussed further?

We acknowledge the fact that separating natural from induced fractures on image logs is far from trivial (line 243-244), but with the set of rules described between line 238-247, we do the best we can with the current dataset. It is outside the scope of the current MS to investigate uncertainty/errors related to the interpretation of the BHI itself but is clearly an important aspect.

281-285 This is confusing.

With this paragraph (now line 278-308), we place our findings in the context of previous studies that specifically target to reconstruct the deformation history of the Parmelan and Jura. We do this to show that our paleostress findings are largely in accordance with previous studies, but there are some differences, which we relate to the aim of the methods uses. In our case, it is adding value to BHI interpretation, rather understanding the full deformation history of the outcrop.

285 'On the contrary' > 'in contrast'

**Corrected.**

303 I assume that by 'the only way' you mean given the type of data that has been collected to date? Maybe instead 'a practical, widely used, and relatively inexpensive way'? But one with several important drawbacks.

Accepted. This line is changed into: 'Borehole images are a practical, widely used, and relatively inexpensive way to sample and characterize the sub-seismic discontinuity network in the subsurface' (line 333-334), followed by some import drawbacks: 'However, there are two main drawbacks (...)' (line 334-338).

304 Maybe start the line with 'In the subsurface of the Geneva basin...' to make it clear that this is a location specific issue.

In this paragraph, we aim to generalize the implications of the methodology, beyond the Geneva Basin only. Therefore, we changed the first sentence of the paragraph into: 'DAs can complement BHI interpretation by providing the discontinuity type of identified background features.' (line 339).

307-321 In 307 you say that image log bias has rarely been investigated, but this isn't really the case, although I guess it depends on what you mean by 'bias'. There have been many studies of the capabilities and limitations of image logs. Bias is a systematic distortion of a result due to some factor. Unless you mean the bias of a specific analyst, the problem is one of inherent ambiguity rather than bias. The kind of reproducible rules, such as in the Andrews et al. reference, are good. But excellent discrimination rules were worked out in the 1990s based on wells with both image logs and core; these are the basis for commercial log picks. There have been many core-to-log comparisons published since 1988 and they mostly come to the same sad conclusion that there is a lot of inherent ambiguity in this aspect of image log interpretation. The reason for this is that many features on image logs look alike. Drilling induced fractures may not have the characteristic shapes and distributions that would allow rules to reliably differentiate them, mineral deposits in natural fractures can be microns thin and undetectable on image logs, and in some case in core inspection. Or fill in sealed fractures can be eroded out. Open natural fractures are not necessarily aligned with current day SHmax. The problem of correctly differentiating drilling and natural fractures or open and sealed fractures has been the focus of several studies since the late 1980s. This section of text can probably be reduced to a short paragraph.

The point I guess is that the image logs are widely used but have mostly intractable limitations, so the kind of outcrop inferences and guidance for log interpretation you provide can be helpful in trying to get reliable data from the logs. Your discussion ought to talk about how general your guidance might be or is it specific to this unit or rock type in this basin.

Good point. We restructured the paragraph to highlight the two ways that DAs improve the BHI interpretation. The first is that discontinuity type can be added to the BHI interpretation. We elaborate on the importance of this in line (339-325). Secondly, DAs can be used to decrease the impact of subjective bias (line 353-359).

On top of this, we are defining how many of the features observed in the well should be extrapolated as background discontinuities; this is quite different from 'guiding the interpretation of single features'. This is discussed in section 6.4 (see line 360-391)

316-320 This section of text describes an important contribution of this MS. But the message seems buried. A clearer description is needed.

Accepted. By restructuring this paragraph as mentioned above, this message is clearer (line 332-359).

327 'barren' and 'mode I' are not equivalent things. And image logs cannot tell if a fracture is barren or not. The mineral deposit veneers on some natural fractures are microns thin and require an SEM to detect, so they (and even thicker deposits) are invisible to current image log technology.

Accepted. 'barren' and 'mode I' are replaced by 'opening mode' (line 340).

A point that I don't see considered is that the outcrop images you show seem to be mostly sealed fractures. Are these fractures filled with calcite deposits (the Results ought to describe this). If the fractures (or at least some of them) in outcrop are calcite filled, that at least is some evidence they are not near surface features but are representative of subsurface deformation. Do you mention this? And if they are sealed, how do they contribute to fluid flow? Or show up as open on image logs? If sealed, is their main role as weaknesses for reactivation during stimulation (Cao et al. point to this as a major uncertainty)? Earlier in the text you mention dissolution along fractures in this basin. Is this an issue worth discussing?

This is a good point, and an additional subparagraph is added to address this topic (line 310-331). In this study, mineral infill and aperture are not extrapolated from outcrop to subsurface, as this indeed requires more work (i.e. absolute timing of fracturing and diagenetic history comparison of outcrop vs. reservoir), but it is important when eventually predicting flow behaviour of the reservoir. This is now discussed.

323-341 I agree with the points here, but this section of text could use some work for clarity.

This paragraph is restructured (line 339-325):

'DAs can complement BHI interpretation by providing the discontinuity type of identified background features. Typically, discontinuity sets defined on BHI (in particular when cores are not available) are all considered as opening-mode fractures. Based on this assumption, a classical workflow consists of defining fracture sets, extracting statistical distributions for these sets, and stochastically extrapolating these distributions at the reservoir scale in a

discrete fracture network model (e.g. Hosseinzadeh et al., 2023). However, the type of discontinuity will impact the evaluation of the flow behaviour of the network in multiple ways. Several studies have demonstrated that stylolites can be either flow conductive or form flow barriers and could potentially induce compartmentalization in subsurface reservoirs (Heap et al., 2014; Koehn et al., 2016). Hooker et al. (2012) and Lander and Laubach (2015) showed that opening fractures are good flow conductors if cement bridges create a natural propping mechanism in the fracture. Finally, the roughness of a discontinuity, which is related to the type, has an impact on its capacity to be reactivated under present-day stress field, which in turn influences its hydraulic aperture under reservoir conditions (Bisdom et al., 2016). These authors also add that typically, shear fractures have a higher roughness than opening fractures, therefore highlighting the importance of being able to constrain fracture type in the reservoir. The DA-methodology provides a prediction of the discontinuity type in the borehole, when the resolution of the BHI is too low to determine this, and there is no core available to correlate the BHI with.

340 (section of Discussion). I think this section ought to be condensed such that it focuses of issues you cover in your Results.

Accepted. We have reduced and rewritten this section and made it less technical (line 360-391):

The tectonic driver of the background network is fundamentally different from the rest of the network, and therefore isolating the background network in the reservoir will improve fracture modeling on reservoir scale. Maerten et al. (2016) developed a method that links discontinuities observed in the well with seismic-scale faults. A given number of random far-field stress states are simulated around the faults, and the perturbation of the stress directions around the faults is calculated. For each simulated stress state, the number of small-scale discontinuities whose orientation fits within the modeled stress field is counted (goodness of fit). The stress state with the highest number of fitting discontinuities is considered the best stress regime, and the discontinuities falling outside this model are discarded from the dataset. The input data for these models are generally all the fractures interpreted from wells, or, in other words, it is assumed that all subsurface fractures are fault-related. Instead, we propose to first isolate the background network, as these discontinuities should be extrapolated to the entire reservoir. Only after this separation, the goodness of fit of fault-related discontinuities should be considered. In this way, the geological understanding of discontinuity formation is better incorporated in the fracture modeling in the reservoir.

Another way how the DA method can improve fracture modeling in the reservoir is in the up-scaling strategy. Berre et al. (2019) advocated for mixing explicit and implicit

representation of fractures in the model as an effective up-scaling method, as it balances accuracy of the process whilst preserving the geometrical complexity. Typically, the selection criterion between implicit and explicit representation is the length of the fractures (Lee et al., 2001). As an alternative to this method, we propose to use the genetic origin of the fracture as a second criterion. Due to its regional character, the background network is very suitable for up-scaling strategies. By assessing the impact of the background network on the effective permeability on reservoir scale, either by analyzing the topology of the network (e.g. Sanderson and Nixon, 2015; Hardebol et al., 2015), or by numerically simulating flow through stochastically generated DFNs (e.g. Agbaje et al., 2023; Kamel Targhi et al., 2025), the decision can be made to either represent the background explicitly or implicitly in reservoir scale models. After the significance of the background network is defined, the next step is to include the discontinuities observed in the well that could not be placed in the framework of the background network. These discontinuities are thus likely created by local drivers and scale differently on the reservoir scale than the background network. For example, if there are seismic-scale faults present in the subsurface, the above mentioned method of Maerten et al. (2016) is a suitable approach to extrapolate these discontinuities to the reservoir scale.

This dynamic workflow will de-risk future geothermal drilling projects in different ways. The separate modeling of the permeability of the background network can be used to assess whether the background only can already produce economically viable fluid volumes, or if seismic-scale discontinuities are essential for production. Also, the well-placing strategy can be adjusted to the heterogeneity of the background permeability field. For example, in the Geneva Basin, most of the background discontinuities are striking NE-SW, and thus, a higher permeability in that direction is expected. A deviation of the well perpendicular to this strike will therefore likely optimize the well screen and thus the fluid inflow.'

360 If you include the effects of fracture abundance in your Results or geological background you should describe what porosity and permeability the host rock has. If host-rock permeability is appreciable then closely spaced fractures (if open) could affect overall permeability due to flow through the host rock between fractures (Philip et al. 2005, SPE Res. Eval. Eng.) If the host rock is impermeable, but the open fractures are not interconnected then the closeness of the fractures to each other should matter. There is a large literature on connectivity and flow (e.g. Long and Witherspoon 1985). Connectivity is not necessarily a function of fracture abundance. But you don't describe connectivity in your outcrop description. Maybe the best move is to make this entire section much shorter and just say that once you have established that the outcrops are representative of the subsurface with your outcrop to image log comparison, you could go back to the outcrops to get this other information that would be useful for modeling.

The primary porosity and permeability of the target reservoir are added in the Introduction (line 66-67), and are very low.

We added this paragraph to the discussion, because we think that it is important to better integrate geological knowledge with fracture modelling in the subsurface. To illustrate the importance of this, we present a guideline how the geological understanding of the network (with the aid of DAs) can be used to improve the workflow for modelling.

363 You mention 'saturation' without putting this concept into context. Maybe best to just leave it out. Where in your Results is there evidence one way or the other to argue for some degree of saturation?

Accepeted. Saturation is removed this rewritten section.

396 The conclusion "Outcrop study is a time and cost-efficient method to obtain a first-order evaluation of the contribution of the background network in the subsurface". I'm sure that this is a true statement. But you have not done a time or cost analysis or a value of information assessment, so I question whether this is a valid conclusion. Maybe the remark belongs at the end of the Discussion along with some ballpark estimates of costs and time of field data acquisition and the potential value of improved image log interpretation. For an example of this and a spreadsheet that can be used to make your calculation, see: Almansour et al. 2020. Value of Information analysis of a fracture prediction method. SPE Reservoir Evaluation & Engineering, 23 (3), 811-823. doi: 10.2118/198906-PA.

Accepted. Indeed, we make no claims about the economics of the method, so refrain to mention this in the conclusions. This point is now changed into: 'Outcrop studies may provide a first-order evaluation of the contribution to flow of the background network in the subsurface'. (line 405-406).

Check the figure captions for the word 'legenda'; should be 'legend'.

**Corrected.**

The titles in the reference list are formatted inconsistently.

Some reference mentioned in the review

Elliott, S.J., Forstner, S.R., Wang, Q., Corrêa, R., Shakiba, M., Fulcher, S.A., Hebel, N.J., Lee, B.T., Tirmizi, S.T., Hooker, J.N., Fall, A., Olson, J.E., Laubach, S.E., 2025. Diagenesis is key to unlocking outcrop fracture data suitable for quantitative extrapolation to geothermal targets. Frontiers in Earth Science 13, 1545052.

Rysak, B.R., Gale, J.F.W., Laubach, S.E., Ferrill, D.A., Olson, J.E., 2022. Mechanisms for the generation of complex fracture networks: observations from slant core, analog models, and outcrop. Frontiers in Earth Science, v. 10, Section Geohazards and Georisks. In Li, Y, Rutter, E.H., Shang, J. and Ji, Y., Eds., Special Issue, Recent Advances in Mechanics and Physics of Rock Fractures across Scales. doi.org/10.3389/feart.2022.848012

---

## Author Comment (AC2)

The manuscript by Hupkes et al. proposes a new approach of using kinematically meaningful fracture and stylolite associations in various analog outcrops to identify regionally persistent patterns of fractures that can be used to guide image log interpretation in wells and are relevant to geothermal exploitation in the subsurface. This is in principle an interesting topic that deserves publication but the manuscript requires to me significant revision before being possibly further considered for publication.

My comments below :

*Title : the first sentence of the title should be removed. They are plenty of examples of occurrence of fracture populations without any stylolites owing to lithology for instance. I do not mean at all I disagree with the idea of working on stylolite-fracture associations, just that the first sentence clearly designed to appealing purpose is wrong in many instances.

*The title has been changed into: "A Fracture Rarely Comes Alone: Associations of Fractures and Stylolites in Analogue Outcrops Improve Borehole Image Interpretations of Fractured Carbonate Geothermal Reservoirs"*

*The Introduction mixes different aspects in a sometimes non logical order, each of which requiring more attention as well as support by references beyond very general considerations. Overall, some reorganization is needed to clearly state the scientific question at hand and to clarify the aims of the study.

*Introduction has undergone major revision. It has the following structure, to make the statements of this study clear in the Introduction:*

- *Understanding discontinuity networks is important for geothermal exploitation of fractured reservoirs.*

- *Borehole images are an important tool for subsurface characterization of discontinuities but come with limitations. Note that we present this already in the second paragraph to underline that in the end, it are the discontinuities in the subsurface that matter.*

- *Analogue outcrops can be used to overcome certain limitations of BHI. To be a good analogue, the outcrop must meet several criteria, among which the shared tectonic/stress history.*

- *To understand the genetic origin of a discontinuity is essential for extrapolation in the subsurface to reservoir scale.*

- *There are several methods to determine paleostress. One is to group discontinuities into associations, following Hancock (1985), and despite its simplicity, it is barely used to link outcrop with subsurface.*

- *Statement of claims. ~45% of the features identified on BHI of the target reservoir in the Geneva Basin can be placed in the predicted framework of the background network.*

*Also, the position of the Geological Background is switched with the Methodology, to improve the readability in the first part of the manuscript.*

*L24 : Addressing the role of discontinuities in forming preferred drains or barriers should also rely on papers from outside the academic circle from the authors (e.g., from host rocks different from carbonate rocks, and mineralizations of other types than calcite; see for instance Grare et al., 2018, Minerals, among others). Also, the respective importance of diffuse fracture networks on rock permeability and fluid flow compared to seismic-scale faults has already been addressed (e.g., Beaudoin et al., 2013; Smith et al., 2022).

*The references are added (line 20 and line 40-41). Indeed, there are many papers who describe potential impact of fractures on flow (either barriers or conducts), and we aim to refer to papers who demonstrate this in the subsurface, either in carbonate or siliciclastic reservoirs.*

* L29-30 : The authors use orientation and relative timing information derived from observations of geometric relationships to gather faults, opening-mode fractures and stylolites into kinematically or mechanically consistent patterns over a wide area in a fold-and-thrust belt and its foreland basin owing to their development in flat-lying strata before folding (e.g., during the layer-parallel shortening stage, see Tavani et al., 2015; Lacombe et al., 2021). In contrast, the authors seemingly consider that fractures related to fold development are only of local significance. This may be true for fractures formed in response to strata bending at fold hinges, but some mesostructures observed in folded strata may be unrelated to the contractional event that caused folding, instead having originated in response to strata burial (Lamarche et al., 2012; Lavenu et al., 2013), foreland flexure (Mercuri et al., 2022), strata exhumation (Bellahsen et al., 2006a), extensional collapse of fold-thrust systems (Tavani et al., 2012) or to other pre-or post-folding tectonic events (e.g. Bergbauer and Pollard, 2004), and some may be of regional significance. The appraisal of the local vs regional significance of fractures therefore may be scale-dependent and should be dealt with more caution eventhough I understand the wish of the authors to simplify the topic.

*We use orientation and discontinuity type to define discontinuity sets, and group these into associations if they are mechanically consistent (i.e. they might have formed in the same stress field). We prefer to refrain from using relative timing to define discontinuity sets, as we believe cross-cutting and abutment relations are often ambiguous, and contain little meaning if the two discontinuities formed in the same stress field that might have prevailed for several millions of years.*

*Regarding the timing of the formation of discontinuity associations with respect to the tilting/bending of the strata, we do not limit ourselves to only the layer-parallel-shortening phase, but to all deformation phases that occurred prior to this tilting and/or bending of the strata. With the methodology we propose, all pre-folding deformation events that fulfill two conditions will be captured: 1) at least two sets of discontinuities are formed that together fit in a discontinuity associations, and 2) the paleostress orientation must be consistent on a regional scale, i.e. on all sides of the basin which contains the target reservoir (see Methodology, line 144-146). The latter criterion defines for us what we mean with a 'regional' scale – i.e. on all sides of the basin that contains the target reservoir.*

*Among the so-called local drivers of fracture development, faults deserve particular attention owing to the perturbations of the regional stress field they cause. The authors should have a close look to the works by e.g., Rispoli, 1981; Rawnsley et al., 1992; Homberg et al., 1997, 2004; Reiter et al., 2024. In particular, the works by Homberg et al. 1997 and 2004 are of prime interest to the study and deserve attention since they document stress perturbations and associated fracture patterns in the vicinity of major left-lateral strike-slip faults in the Jura (Pontarlier and Morez faults), keeping in mind that the area investigated is close to a  very similar structure, the Vuache fault, which, on top of that is still seismically active. Again, this questions the scale at which one should consider the fracture patterns to be of local or regional significance in the kind of study carried out by the authors.

*In this study, we focus on predicting the background network (i.e. diffuse discontinuities) in the subsurface, rather fault-related discontinuities. We by no means claim that the background network constitutes the complete network in the subsurface - moreover, we do not know if the contribution of the background network to flow in the target reservoirs is significant. It is more than likely that faults and discontinuities related to the stress perturbation around the faults influence locally the flow characteristics of the reservoir, but that is outside the scope of the current study. Interestingly enough, we do find discontinuity associations that are similar in orientation to those observed in the other outcrops (Parmelan, Jura) network close to the seismically active Vuache fault (see stations 32, 35 and 39 on figure 10).  Therefore, we define them as part of the background network, that*

*formed prior to tilting of the strata. So even though the complex history of this fault, with several reactivation phases, the regionally consistent background network is still present within several 10s of meters away from the Vuache fault.*

*L28 and 83-84 : The definition of a set of fractures by the authors is a bit simplistic and should be based instead on a more complete set of criteria, including a common orientation in either raw or unfolded attitude, common relative chronology with respect to other mesostructures and shared mode of deformation (e.g., Pollard and Aydin; Bellahsen et al., 2006a; Ahmadhadi et al., 2008; Lacombe et al., 2011; Sanderson et al., 2024). Due credit should be better given to such existing literature, including the statistics behind defining distinctive fracture sets.

*We used common orientation and mode of deformation to define the discontinuity sets that make up the associations. In particular the discontinuity mode is essential, as this is used for defining the discontinuity associations. It should be noted that in the end we are not interested in single discontinuity sets, but in the stress field in which they formed, because that is what we need to extrapolate discontinuities to the subsurface. To be more consistent on this point, in the Introduction, the definition of sets is replaced by a focus on the tectonic driver of discontinuities (line 40-50 ). In the Methodology, references for the definition of sets are added (Lacombe et al. 2011, Sanderson et al. 2024, see line 113).  As mentioned above, relative timing is of subordinate importance for defining discontinuity sets in our study, as we focus on associations of discontinuities and the stress field in which they are formed, rather than individual sets.*

*L29-31 : again, references should be provided from out of the academic circle of the authors : e.g., Bergbauer and Pollard, 2004; Casini et al., 2011;  among others

*These are indeed good references and added to the manuscript (line 42-43).*

*L55 : That multiple discontinuity sets can form coeval in a single stress field is now nicely supported by absolute dating of syn-tectonic calcite mineralizations, which allows for a  better appraisal of the local vs regional tectonic significance of fracture networks (e.g, Beaudoin et al., 2018; Parrish et al., 2018; Zeboudj et al., 2025). However, in some instances, the use of geochronology may either simplify or sometimes increase the apparent complexity of the fracture network analyzed on the sole basis of orientation and kinematic criteria. Geochronology may reveal very similar ages for veins of significantly contrasting orientations, which questions the perfect stability of the direction of the maximum principal stress during a fracturing event and the range of orientations allowed to define a fracture set considering possible stress variation at the scale of the strata or close

to faults, in a time span lower than the few My of uncertainties of the geochronology technique. Conversely, absolute dating can, in some cases, complicate the interpretation of deformation history particularly when multiple distinct ages are obtained from fractures belonging to the same set defined on the qualitative criteria above. I would suggest much more caution with the hypothesis of constant principal stresses at the regional scale (L119).

*We do not assume there is a regional consistent regional stress field, but the results show that in fact there is consistency with respect to orientation of σ1 that formed the discontinuity associations of the background network (see figure 10).*

*Geochronology in the form of dating calcite veins is indeed a very interesting way to constrain timing of the formation of the background network. It will give a minimum age constraint on the forming of the discontinuities, as the timing of calcite precipitation does not necessarily coincide with the propagation of a fracture. On top of this, we don't know how much time is involved in the creation of the background network. If we consider that they are formed under sub-critical conditions (e.g. numerical work on sub-critical crack growth of Welch et al. 2019, Olson 1993, 2004, 2007), it might have taken 10s Ma. Therefore, it is expected that different ages will come out for the discontinuity association. But in the current study, we don't have any absolute time constraints on the formation of DAs. That is why we focus on relative timing with respect to tilting/bending of the strata.*

*L107-110 : the authors should be more careful when using the Andersonian stress hypothesis according to which one principal stress is generally vertical, especially in fold-and-thrust belts. As stated in the discussion paper by Lacombe, 2012 (section 4), mesostructures that yield one stress axis perpendicular to bedding while the other two lie within the bedding plane (e.g., bed-perpendicular joints/veins or stylolites, or Layer-Parallel Shortening (LPS) – related microfaults) yield a sub-vertical paleostress axis only after backtilting to their prefolding attitude (unfolding), which in turn implicitly leads to consider them as pre- or early-folding. However, it has been argued that such structures could have developed within tilted layers, hence possibly under a non vertical principal stress, if bedding anisotropy was able to significantly reorient stresses or if flexural slip occurred at very low friction so that the principal stresses rotated but remained either parallel or perpendicular to bedding (e.g., Tavani et al., 2015). A kind of circular reasoning may thus be involved when chronology is based on an Andersonian assumption only. As a result, the classical fold test must be preferred.

*It is indeed an assumption that DAs are formed in Andersonian stress field, as mentioned in the Methodology section (line 136). It is outside of the scope of the current manuscript to quantify the orientation of the bedding at the timing of fracturing. We consider the simplest*

*solution the most plausible, i.e. they formed in Andersonian stress fields. In the Discussion, we discuss the possibility that the DAs are related to LPS (line 286-300). LPS will likely produce associations of discontinuities of reverse regime with a principal stress close to vertical to the bedding. However, for making the link with the subsurface, it is crucial to identify that these discontinuities formed prior to tilting of strata and related strain localization. That allows us to predict the presence of these structures in the subsurface of the Geneva Basin.*

*L292 : I suggest to change into : … predates the onset of localized deformations occurring during fold growth and late stage fold tightening.

*Accepted. See line 295-297.*

*Stylolites are expected to be important players according to the title of the manuscript and the approach is supposed to use fracture and stylolite relations, but the 'discontinuity' terminology used throughout the manuscript leads to overlook the role of stylolites. Poor use is made of stylolites and the most recent literature on stylolites is properly ignored (eg. Toussaint et al., 2018). Especially, it has been shown that the roughness of a stylolite brings some signal that can be treated and used to derive the state of stress prevailing at the time the stylolite ended its development. This applies to compaction-related sedimentary stylolites as well as to tectonic stylolites. In contrast to sedimentary bedding-parallel stylolites used to derive paleodepth constraints (eg., Beaudoin et al., 2019; 2020; see also the work by a senior author of the manuscript), tectonic stylolites the manuscript seemingly focuses on can bring very useful kinematic information for fracture analysis, beyond the classical papers (e.g., Marshak and Engelder). Tectonic stylolites can be either compressional or strike-slip in type (e.g., Beaudoin et al., 2016, 2020). In this case, the accurate interpretation of the stylolites in terms of stress is highly relevant and potentially predictive in fracture network studies. Adding one or two sentences on this topic may strengthen the 'stylolite' aspects of the manuscript which otherwise remains weak and nearly useless as is.

*We added the references for stylolite as paleostress marker in the Introduction (Beaudoin et al. 2016, Toussaint et al. 2018, see line 53). Although it is a highly interesting topic, using stylolites to reconstruct magnitudes of paleostress is outside of the scope of the current manuscript, and therefore references are not included. It would indeed be an interesting next step to complement the current work with paleostress magnitudes, and compare if all outcrops experienced similar stress magnitude. However, for tectonic stylolites, this may*

*be challenging, as the assumption that σ2 and σ3 are of same magnitude is no longer valid, as for bed-parallel stylolites (see Ebner et al. 2010, JGR Solid Earth).*

*In the Results section, when defining the regional events, we appreciate the fact that bed-perpendicular stylolites are indicative of a strike-slip or reverse regime (line 210-220).*

*More in general, the term discontinuity in this manuscript is chosen to include both fractures and stylolites. The point of this study is that discontinuities, being it fractures or stylolites, should be grouped based on the stress field in which they formed, instead of interpreting isolated features or sets, if to have added value for subsurface characterizing and extrapolation. Grouping of discontinuities is also important in the absence of stylolites: it makes no sense to treat a conjugated pair of shear fractures as two independent fracture sets belonging to different events when populating a fracture model. It can be seen in figure 8 that in some stations, tectonic stylolites are not used for defining an association.*

\*The role of fractures in fluid flow in unclear, especially the role of diffuse fracture networks compared to major drains (eg, seismic scale faults), see for instance Beaudoin et al., 2013. Also, the authors discuss in which situations fractures may enhance fluid flow, but all the field examples document sealed mode I fractures. In the vast majority of studies as well as in modeling works, the features conducting fluid are commonly assumed to be open fractures (e.g., Wennberg et al., 2016). Would the authors expect that in the subsurface, the fractures of interest (i.e., conductive) be open or only partly sealed while those in the outcrops are filled with calcite ? Or do they consider that already sealed fractures may be re-opened and made conductive under the current stress field ? It is unclear since authors speak about either mode I fractures or open fractures. It should be kept in mind that mineralization of mode I fractures and faults requires some specific chemo-physical conditions (see the synthesis by Laubach et al, 2019) unlikely to be met very close to the surface. Open fractures are more likely to be found close to the surface than under a certain overburden. In other words, the structural, burial and fluid flow history of exposed rocks and rocks in the subsurface may differ. This point requires an even brief discussion.

*Accepted. A section in the Discussion is dedicated to this point (line 310-331):'*

*'6.2 Infill and aperture of discontinuities*

*The DA-method can be used to predict the geometry of the background network in the target reservoir, but is limited in extrapolating the aperture and mineral infill of fractures. The geometry is useful when considering stimulating the reservoir, as even the sealed discontinuities may create a strength anisotropy that will control the orientation and propagation of hydraulic fractures (Cao and Sharma, 2022; Rysak et al., 2022). However, for*

*predicting flow behaviour in the reservoir caused by natural discontinuities, modeling the aperture and mineral infill of discontinuities is crucial, as only (partially) open discontinuities might contribute to the flow. At the same time, outcrops should be treated with care when extrapolating these properties to the subsurface (e.g. Bauer et al., 2017; Peacock et al., 2022), also when the link between outcrop and subsurface is established with the DA-method. The timing of fracturing, emplacement of the infill and potential dissolution are important factors to consider when extrapolating these characteristics to the subsurface. On the Parmelan, for example, many small-scale (<10 meter) fractures of E1 and E2 are calcite filled (e.g. see figure 3). The diagenetic evolution can be used to constrain the timing of calcite cement formation in the outcrop (e.g. Lavenu and Lamarche, 2018; La Bruna et al., 2020), and subsequently provide insights how the aperture of these discontinuities can be modeled in the subsurface (Elliott et al., 2025). On the other hand, the large-scale fractures (> 100 m) of E1 on the plateau are currently conductive due to dissolution and karstification (see figure 4). It depends on the timing of fracturing and subsequent dissolution if the conductivity of these fractures can be used as an analogue for the paleokarst network that is observed on top of the Lower Cretaceous in the subsurface of the Geneva Basin (Eruteya et al., 2024). If E1 was formed prior to sub-aerial exposure of the Lower Cretaceous during the Paleogene, it is likely that they partially controlled the orientation of karst development. On the contrary, if the karstification on the Parmelan only occurred after the exhumation in the Pliocene, similarly dissolved fractures cannot be expected in the subsurface. So, in order to predict the aperture and if discontinuities are sealed in the reservoir, solely based on outcrops, the timing of fracturing and the diagenetic evolution of the formation are both essential to predict which discontinuity sets in the subsurface are likely to be conductive. Another possibility is to use borehole data to assess which discontinuities are conductive, and the DA-method can be part of the workflow to improve the interpretation.'*

*The reference of Beaudoin et al. 2013 is added in the Introduction (line 40) as well as Laubach et al. 2019 (line 32).*

*Like the physical properties (e.g., permeability) of fault rocks from exhumed ancient faults may hardly be extrapolated to the in situ hydraulic behavior of deep active faults because during uplift these rocks underwent unloading, decrease in temperature and changing fluid-rock interactions (including weathering), one can wonder whether fracture networks identified in exposed rocks are reliable analogs for the mechanical and hydraulic properties of subsurface fracture networks despite similar regional tectonic history and host rocks.

*We added a section in the Discussion to touch upon this (line 310-331, also see above). In general, the 'goodness' of the analogy of the outcrop for the subsurface depends on what it*

*is used for. The DA-method is aimed to predict the geometry of the background network, and in this study we show why the outcrops we are used are a good analogue for the subsurface of the Geneva Basin (pre-tilting discontinuity associations that are regionally consistent in orientation). Extrapolating discontinuity aperture and infill from the outcrop to subsurface to assess hydraulic properties of the reservoir remains challenging (see e.g. Bauer et al. 2017, Peacock et al. 2022), but in the discussion we illustrate how the DA-method might contribute to this (line 361-385).*

*Since veins contain nearly pure calcite material compared to the host rock, one could expect some kind of strength change in the rock due the structural diagenesis, with the vein network causing possibly some strength anisotropy at the scale of the reservoir. Would it be an alternate parameter to usefully consider during exploitation ?

*Yes, this is true, in particular when stimulating the reservoir with hydraulic fracturing is considered. It is added in the Introduction (line 21-22):*

*'Also, natural discontinuities impact the rock strength, which is essential to consider in the case of hydraulic stimulation of a reservoir (Cao and Sharma, 2022; Rysak et al., 2022).*

*and the Discussion (line 310-313):*

*'The DA-method can be used to predict the geometry of the background network in the target reservoir, but is limited in extrapolating the aperture and mineral infill of fractures. The geometry is useful when considering stimulating the reservoir, as even the sealed discontinuities may create a strength anisotropy that will control the orientation and propagation of hydraulic fractures (Cao and Sharma, 2022; Rysak et al., 2022).'*

*What about stylolites in fluid flow studies ? The authors report that stylolites may behave as drains or barriers. First, again, citing only Bruna et al. is not a fair acknowledgement of previous work on the topic (see for instance, Koehn et al., 2016; Heap et al., 2014; Braithwaite, 1989). In addition, when speaking about compartmentalization of the reservoir in term of fluid flow, I guess the authors also consider sedimentary, i.e., non-tectonic bedding-parallel stylolites, the development of which has nothing to do with the topic of the manuscript. It may be misleading to mix features as different as fractures and stylolites, or tectonic and non-tectonic features.

*The useful references are added for the stylolites are added (line 345). In general, flow modeling is outside of the scope of the current manuscript. We only predict geometry of the background network in the subsurface and validate this with BHI-data. In the*

*discussion, we do mention how our proposed methodology might improve fracture modeling and flow simulation workflows.*

*Bed parallel stylolites are indeed an interesting feature (we prefer this over 'non-tectonic' stylolites: they are as indicative for paleostress orientation and thus the tectonic regime as bed perpendicular stylolites). They are not included in the presented DAs, as there were no other discontinuity sets observed that are in association with these stylolites, and therefore they did not pass the criterion that we need a minimum of two discontinuity sets before defining an association. Nonetheless, we do think they will have an impact on flow in the reservoir. However, the outcrop is of limited help to predict the impact on the flow of these stylolites in the subsurface: in the outcrop, most bed-parallel-stylolites are open and would act as flow conduits. However, this might well be related to the fact that there is no overburden in combination with recent dissolution, which might not be expected in the subsurface in the target reservoir.*

*If I understand well, the authors show that in the area investigated, ~40-50% of the fractures identified by borehole imaging correspond to natural fracture patterns, and discard the uncorrelated fractures as being more recent structural objects (L 321). The authors do not explain which process the formation of these uncorrelated fractures can be ascribed to, or if (why?) they necessarily postdate the development of the regionally consistent fracture sets. In addition, should they predate or postdate the formation of the natural fracture sets, I would expect some of the uncorrelated fractures to be of possible interest by enhancing connectivity of the regional natural fracture sets despite being of local significance, hence permeability of the reservoir . This would deserve a bit more attention for the interested reader.

*Indeed, we focus only on the discontinuities that we can place into regional consistent DAs, and refrain from making claims about the other discontinuities that do not fit in this concept, as this would be speculation. The fractures observed in BHI that do not fit within the predicted DAs could either be formed before or after the 45-50% we define as background network, we do not know. Of course, for fluid flow, they are important, because they are present. But they explain these fractures and assess their importance for the reservoir in the Geneva Basin is outside of the scope of the current paper, but clearly should be addressed when executing flow simulations of the reservoir.*

* I think that accepting a maximum deviation of ~30° for azimuth and dip of fractures recognized on the BHI is reasonable, but there is no justification for this acceptability 'threshold'. Is it purely arbitrary and does it rely on observations or established statistics ? Please justify.

*There is uncertainty related to the orientation of the picked fractures. On top of that, in the outcrop, we also observe variability in the orientation of discontinuities that we grouped into DAs.*

*Therefore, we consider a 30 degree deviation (azimuth + dip) as acceptable. A higher deviation might even be reasonable, but we do not exceed the 30 degrees, as the individual sets that make a single DA would start to overlap. This justification is added in line (268-271).*

\* What about the interpretation of the clockwise rotation with depth of the dip direction of bedding planes in GEO-01 ? The authors relates it to the probable presence of a fold, itself potentially related to a fault, at ~480m depth. First, the authors should substantiate this interpretation and possibly discuss the significance of this fold (geometry, consistency with regional features). Second, following the authors' logic, this fold and associated fault have expectedly caused localized brittle deformation, for instance in the form of extensional fractures at fold hinge. Did the authors try to incorporate this aspect in their interpretation / scenario? How do they explain that the % of uncorrelated fractures is similar with Geo-02 ? Third, several studies have shown that stress may be perturbed close to the tip of a reactivated / propagating fault even in still flat-lying strata, causing local directional perturbation of both stress orientation and magnitudes during LPS which may strongly influence the distribution of fractures in advance of fold development (e.g., Bellahsen et al., 2006b; Amrouch et al., 2010). I would expect the authors not to simply brush aside this point. This comment is also relevant to fractures formed in the vicinity of the Vuache fault which formed prior to the Alpine LPS (Oligocene extension, Homberg et al., 2002) and hence was prone to reactivation during the subsequent Alpine compression.

*We did not focus on the fold and/or fault that might be present in the well, because that is not the scope of the current manuscript. We focus on characterizing the background network based on outcrop observations. It is by no means the aim to present a 'complete' interpretation of the BHI that explains all features observed in the well.*

*It is an interesting observation that the % of uncorrelated fractures is similar in Geo-01 and Geo-02. However, as in this study we focus on the background network, we prefer to turn the observation around: the same percentage of features in both wells can be related to the predicted background network. This underlines the regional character of the background network, or in other words, that this portion of the fractures may be extrapolated to the entire reservoir. As mentioned above, it was outside of the scope of the current study to*

*investigate how the features that are not related to the background network compare between the two wells.*

*In this study, we document DAs in the field. It turns out there is a regional consistency in the orientation of DAs, regardless of the distance to regional faults such as the Vuache. We acknowledge that local stress perturbation in flat-lying strata may occur, but with the methodology presented in the current MS, they would not be documented, as they don't reveal a regional pattern. Yet we manage to explain 40-50% of the subsurface features. As mentioned above, we are aware that local stress perturbations may have created fractures - but they are outside of the targeted discontinuities in this study, as it will require a different approach to use outcrops to predict the stress perturbation-related discontinuities in the subsurface.*

\* The authors make a big deal about the use of directional and kinematic characteristics of fracture –stylolite associations in outcrops, but I am also wondering whether additional information such as fracture dimensions (length, vertical persistence across strata) or fracture connectivity owing to the type of fracture intersections, cross-cutting relationships with stylolites could not be also useful to built a reliable fracture and permeability model of the reservoir since these attributes can hardly be assessed using well bores.

*Yes - length, vertical/lateral variability, clustering, aperture, connectivity are all important factors to consider when building the reservoir model, and these factors are hard to assess with borehole data only. It remains challenging to proof that outcrops can serve as an analogue for these characteristics (see e.g. Bauer et al. 2017, Peacock et al. 2019). As this is an important point, it is added in line 314-316 of the Discussion. We believe that the Parmelan might be a good outcrop to study these characteristics, as it is such a large exposure. The best approach to validate its analogy then would be to conduct flow simulation through networks as observed on the Parmelan, and compare this with the dynamic data of the reservoir. However, such a study is beyond the scope of the current MS.*

\*The authors conclude about the success of their approach in the particular case of the Geneva basin, and encourage people to apply it to other case studies, but what could be the limitations of this approach if applied blindly to a different setting ? I guess the change in lithology with depth, the stress discrepancy between above and below a mechanically weak decoupling layer or stress perturbations caused by large-scale structures could be part of the answer, but the authors should elaborate a little bit on that point.

*A clear limitation of the proposed method, as for most methods that use outcrops as analogue for the subsurface, is the aperture and infill of discontinuities. We added a section in the discussion to elaborate on this (line 310-331). Another limitation is the availability of analogue outcrops in the vicinity of the target reservoir.*

*We think our methodology is not limited to a single lithology type (in this case limestones), as discontinuity associations might form in any rock type. Stress changes with depth are indeed an interesting point, and in many paleostress studies neglected. By considering stress permutations when grouping DAs into events (Methodology, line 138-143), we aimed to include potential stress regime changes that are caused by change in overburden. Mechanically weak decoupling layers might create differences in the regional stress field, but for the proposed DA-methodology, we think it will have minimal impact. The outcrops are of the same age and same lithology as the target reservoir (as in most analogue outcrops studies), and as we focus on the background-forming-events (i.e. prior to fold-and-thrusting and tilting of the strata), we think it is a safe assumption that the target reservoir and analogue outcrops had a similar position with respect to weak layers at the timing of fracturing.*

I hope that these comments will help the authors improve their manuscript.

*Yes, thank you kindly!*

---

## Referee Report (RR1)

I thank the editor for asking me to assess the revised manuscript and revision memo based on his impression that the revision and revision memo may not have been fully responsive to my review. In general, I think the authors have done a thorough revision and that the concerns that I raised have been attended to. There are several technical and usage issues where we do not agree, but the revised text (and the published review comments) should allow readers to make up their own minds about it.

I've highlighted some text from the revision memo that underline the differences of opinion. Point 1 is the main issue; points 2-4 are minor.

1. *"To understand the genetic origin of a discontinuity is essential for extrapolation in the subsurface to reservoir scale."*

*"…we strongly believe that, to improve subsurface discontinuity modeling from a geological perspective, it is essential to consider the stress fields in which discontinuity associations are formed. Without such a step, we cannot predict if a given discontinuity (association) has to be extrapolated to the entire reservoir (with a spatial variability), or only along a fault/fold…"*

*"It is true that we don't know the timing of formation of the fractures and stylolites described in the MS. However, if discontinuities fit in the framework described by Hancock (1985), the simplest interpretation is that they formed in the same stress field. In this workflow, the relative timing between individual discontinuities is of subordinate importance. As we know, the duration of the stress field in which discontinuities form is much longer than the time needed for a fracture to grow and stop, relations between single features do not say much."*

The main issue is the assertion (1) that "To understand the genetic origin of a discontinuity is essential for extrapolation in the subsurface to reservoir scale." It is notably challenging to discover the 'genetic origin' of most fractures, so this seems like a high bar to me. For example, did this particular fracture (or array of fractures) form due to elevated pore pressure or to tectonic shortening/extension? If making those kinds of distinctions was 'essential' then extrapolation would indeed be hard to do (as some of the references cited in the revised MS show). But as I think the text currently makes apparent, the aim in the MS is the less ambitious one of assessing whether groups of fractures and stylolites formed in arrays of regional extent in flat lying rocks or if they are just associated with faults. Although I think the revised text is adequate, my preference would be to make it a bit clearer to the reader what level of 'genetic origin' information is needed and to underline that many actual subsurface fracture arrays (as documented in the literature) are not part of arrays like those shown in figure 2. *If* outcrops contain arrays of fractures and stylolites that can be associated with each other by the model in figure 2, and the arrays are widespread around the margins of a basin, then an extrapolation of the patterns into the basin is defensible, especially if the patterns can account for orientation patterns documented from the subsurface with image logs. But, the relative timing between individual fractures remains valuable information to collect (and is unfortunately rarely documentable in subsurface data sets), and the relative durations of stress fields and fracture growth are assumptions. I don't agree with the revision memo's apparent disparagement of the approach of defining fracture sets, but this point doesn't seem central to the point of this MS.

2. A couple of (mostly) usage issues.

*"We chose the term discontinuity, as we want to include both stylolites and fractures in our proposed methodology. In this study, we do not focus on individual sets, but on the stress fields in which associations of discontinuity are formed…"* and *"As mentioned above, we prefer to refrain from using relative timing for defining sets, as this is not consistent with the DA-methodology. We do infer time relations between different stress fields. These are based on cross-cutting relationships, but we acknowledge that this can be tricky. In this case we do this however, because we are sure we are looking at features from different DAs."* The revision clarifies one of my concerns with the discontinuity association terminology by mentioning 'stylolites and fractures' early in the text. I think readers may still wonder how crosscutting relations can both be used and not be used to make these distinctions.

*"We prefer not to use 'kinematically compatible', as we think this is somewhat misleading: discontinuities are compatible with respect to the information they deliver on the stress field in which they formed, not for their kinematics."* I guess this response might reflect a misunderstanding of my original comment. The faults, opening mode fractures, and stylolites in figure 2 are kinematically compatible in that their movement and configuration are consistent with the stress field indicated there, that's why they can be used to infer a paleostress field.

3. *"…fault has the connotation of large displacement (meters)…"* I don't agree with this assertion, but the term 'shear fracture' is entrenched in the literature despite Pollard and Aydin (1988).

4. As I noted in the initial review, I don't think the word 'analogy' in the Abstract, line 35, and elsewhere, is being used correctly (or at least some other phrase might be clearer). Maybe this is a translation issue. The term 'outcrop analog' (something comparable to another) is widely used but making the inference that the outcrop matches the subsurface is not usually called an analogy. An analogy is a comparison between one thing and some other thing that helps explain or clarify. Usually this is a comparison of two otherwise *unlike* things based on resemblance of a particular aspect, which is not the case in this instance because the authors are claiming that the discontinuity associations in the outcrops are *the same* as those in the subsurface. Calling this proposed correspondence an analogy may hinder comprehension of what the authors are proposing and furthermore this usage hides a claim of the paper. Why not just say that 'based on the similarity of the outcrop patterns to the elements of those patterns that can be discerned on image logs, we infer that the outcrop patterns are representative of the subsurface.' Or in line 35: "However, [demonstrating?] the analogy [correspondence?] between outcrop and subsurface is far from trivial (e.g. Bauer et al., 2017; Peacock et al., 2022). To establish the analogy [correspondence?] between outcrop and subsurface…" The further inference that if two or more things agree with one another in some respects they will probably agree in others is matter for the Discussion (that is, if the orientation patterns match, perhaps the length and connectivity patterns will match). This word usage is certainly not a technical issue worth holding up the MS and I'm happy with whatever the authors and editor decide.

Comments keyed to lines in the text

Since I read through the MS again I've marked a few imperfections or questions.

Line 55: "The concept that multiple discontinuity sets ['discontinuities' instead of 'discontinuity sets'?] can form in a single stress field is largely sensed by structural geologist[s] (e.g. Groshong, 1975)…"

Line 394-5 Don't you mean 'Associations of genetically related discontinuities that form the background network produced by a far-field paleostress are defined in the field…' Otherwise it sounds like you mean fractures created by the current state of stress.

---

## Author Response (AR2)

**Reply to RC1**

*We kindly thank the referee for their comments on the revised manuscript and the revision memo. Below, we give our point-by-point response in italic blue font.*

I've highlighted some text from the revision memo that underline the differences of opinion. Point 1 is the main issue; points 2-4 are minor.

1. *"To understand the genetic origin of a discontinuity is essential for extrapolation in the subsurface to reservoir scale."*

*"...we strongly believe that, to improve subsurface discontinuity modeling from a geological perspective, it is essential to consider the stress fields in which discontinuity associations are formed. Without such a step, we cannot predict if a given discontinuity (association) has to be extrapolated to the entire reservoir (with a spatial variability), or only along a fault/fold..."*

*"It is true that we don't know the timing of formation of the fractures and stylolites described in the MS. However, if discontinuities fit in the framework described by Hancock (1985), the simplest interpretation is that they formed in the same stress field. In this workflow, the relative timing between individual discontinuities is of subordinate importance. As we know, the duration of the stress field in which discontinuities form is much longer than the time needed for a fracture to grow and stop, relations between single features do not say much."*

The main issue is the assertion (1) that "To understand the genetic origin of a discontinuity is essential for extrapolation in the subsurface to reservoir scale." It is notably challenging to discover the 'genetic origin' of most fractures, so this seems like a high bar to me. For example, did this particular fracture (or array of fractures) form due to elevated pore pressure or to tectonic shortening/extension? If making those kinds of distinctions was 'essential' then extrapolation would indeed be hard to do (as some of the references cited in the revised MS show). But as I think the text currently makes apparent, the aim in the MS is the less ambitious one of assessing whether groups of fractures and stylolites formed in arrays of regional extent in flat lying rocks or if they are just associated with faults. Although I think the revised text is adequate, my preference would be to make it a bit clearer to the reader what level of 'genetic origin' information is needed and to underline that many actual subsurface fracture arrays (as documented in the literature) are not part of arrays like those shown in figure 2. *If* outcrops contain arrays of fractures and stylolites that can be associated with each other by the model in figure 2, and the arrays are widespread around the margins of a basin, then an extrapolation of the patterns into the basin is defensible, especially if the patterns can account for orientation patterns documented from the subsurface with image logs. But, the relative timing between individual fractures remains valuable information to collect (and is unfortunately rarely documentable in subsurface data sets), and the relative durations of stress fields and fracture growth are assumptions. I don't agree with the revision memo's apparent disparagement of the approach of defining fracture sets, but this point doesn't seem central to the point of this MS.

*We thank the reviewer for clarifying the issue of using 'genetic origin'. We think the reviewer is correct that the 'level of genetic origin' might be the cause of a misunderstanding.*

*Indeed, the current manuscript makes the claim that for the purpose of developing better subsurface fracture models, it is important to differentiate background/diffuse discontinuities from the fold/fault related discontinuities, as this has direct implications for the distribution of these discontinuities on the reservoir-scale. This is what we mean with 'genetic origin'. To ensure this is clear for the reader, we added this in line 49-50:*

*'Understanding the genetic origin of discontinuities (i.e. background vs. fold/fault related) is therefore essential for extrapolation of discontinuity geometry to reservoir scale.'*

*Regarding the relative timing between individual fractures, we might not have expressed our opinion in the best way. But we believe that within a discontinuity association, the relative timing of individual sets is of secondary importance, as it will not provide additional information on the paleostress field in which the DA has formed. We do acknowledge that cross-cutting relationships may contain relevant information. In particular in the case of veins, where the cement of the younger cross cuts the cement of the older, it is a very clear indication of relative timing. We used this to define the relative timing between E1 and E2, i.e. between different DAs.*

*To clarify this point regarding the relative timing, we added the line (116-118):*

*'Relative timing based on cross-cutting and abutment relationships between individual sets within a discontinuity association are not considered in this study, as they do not reveal additional information on the orientation of the paleo principal stresses.'*

2. A couple of (mostly) usage issues.
*"We chose the term discontinuity, as we want to include both stylolites and fractures in our proposed methodology. In this study, we do not focus on individual sets, but on the stress fields in which associations of discontinuity are formed..."* and *"As mentioned above, we prefer to refrain from using relative timing for defining sets, as this is not consistent with the DA-methodology. We do infer time relations between different stress fields. These are based on cross-cutting relationships, but we acknowledge that this can be tricky. In this case we do this however, because we are sure we are looking at features from different DAs."* The revision clarifies one of my concerns with the discontinuity association terminology by mentioning 'stylolites and fractures' early in the text. I think readers may still wonder how crosscutting relations can both be used and not be used to make these distinctions.

*"We prefer not to use 'kinematically compatible', as we think this is somewhat misleading: discontinuities are compatible with respect to the information they deliver on the stress field in which they formed, not for their kinematics."* I guess this response might reflect a

misunderstanding of my original comment. The faults, opening mode fractures, and stylolites in figure 2 are kinematically compatible in that their movement and configuration are consistent with the stress field indicated there, that's why they can be used to infer a paleostress field.

*We thank the reviewer for the clarification; we believe that we agree that multiple discontinuity sets are compatible in the sense that they deliver consistent information regarding the paleostress field.*

3. *"...fault has the connotation of large displacement (meters)..."* I don't agree with this assertion, but the term 'shear fracture' is entrenched in the literature despite Pollard and Aydin (1988).

*We replaced the term 'shear fracture' by fault throughout the text. To make clear to the reader that we are typically dealing with very small displacements, we added 'small-scale' as a prefix in some occasions.*

4. As I noted in the initial review, I don't think the word 'analogy' in the Abstract, line 35, and elsewhere, is being used correctly (or at least some other phrase might be clearer). Maybe this is a translation issue. The term 'outcrop analog' (something comparable to another) is widely used but making the inference that the outcrop matches the subsurface is not usually called an analogy. An analogy is a comparison between one thing and some other thing that helps explain or clarify. Usually this is a comparison of two otherwise *unlike* things based on resemblance of a particular aspect, which is not the case in this instance because the authors are claiming that the discontinuity associations in the outcrops are *the same* as those in the subsurface. Calling this proposed correspondence an analogy may hinder comprehension of what the authors are proposing and furthermore this usage hides a claim of the paper. Why not just say that 'based on the similarity of the outcrop patterns to the elements of those patterns that can be discerned on image logs, we infer that the outcrop patterns are representative of the subsurface.' Or in line 35: "However, [demonstrating?] the analogy [correspondence?] between outcrop and subsurface is far from trivial (e.g. Bauer et al., 2017; Peacock et al., 2022). To establish the analogy [correspondence?] between outcrop and subsurface..." The further inference that if two or more things agree with one another in some respects they will probably agree in others is matter for the Discussion (that is, if the orientation patterns match, perhaps the length and connectivity patterns will match). This word usage is certainly not a technical issue worth holding up the MS and I'm happy with whatever the authors and editor decide.

*We thank the reviewer for elaborating on the term 'analogy', and we apologize for not addressing this comment properly after the first review. We indeed claim that the geometry (orientation and type) of a part of the network observed in outcrop (namely, the background) is the same for the outcrop and subsurface. To ensure this is clear, the text is adjusted in the abstract in line 1:*

*'In this study, we present a method that uses associations of discontinuity sets to demonstrate similarities between the outcrop and subsurface.'*

*And in line 10-11:*

*'Given the regional character of these events, we predict that the target reservoir is impacted by*

*them as well'.*

*And in line 35-37:*

*'However, demonstrating that the outcrop and subsurface are similar is far from trivial (e.g. Bauer et al. 2017, Peacock et al. 2022). To justify the usage of analogue exposures for characterizing discontinuities in the subsurface…'*

Comments keyed to lines in the text

Since I read through the MS again I've marked a few imperfections or questions.

Line 55: "The concept that multiple discontinuity sets ['discontinuities' instead of 'discontinuity sets'?] can form in a single stress field is largely sensed by structural geologist[s] (e.g. Groshong, 1975)…"

*Accepted (line 55).*

Line 394-5 Don't you mean 'Associations of genetically related discontinuities that form the background network produced by a far-field paleostress are defined in the field…' Otherwise it sounds like you mean fractures created by the current state of stress.

*Correct, it has been adjusted (line 395).*

**Reply to RC2**

*We kindly thank the reviewer for the additional comments on the manuscript. We give our point-by-point response below in blue italic font.*

**Suggestions for revision or reasons for rejection**

(visible to the public if the article is accepted and published)

I have now reviewed the revised version of the manuscript by Hupkes et al. My overall feeling is that the manuscript has improved but still requires minor revision before potential publication.
The authors have addressed most of my comments and responded often convincingly to them. However, I have still some recommendations that should be taken into account before acceptance.

*When talking about fracture associations and more generally about the brittle deformation pattern in fold-and-thrust belts, I think that due credit must be given to the review paper by Tavani et al., 2015, Earth-Science reviews which is highly relevant to the topic of the manuscript, should it be about fracture occurrence and types, associations, chronology with respect to folding and even stress interpretation.

*The reference is added in line 42*

*L79 : Barbier et al. MPG, 2012 should also be considered here

*The reference is added in line 48*

*L83 : should be roughness instead of shape'

*This is adjusted (line 53).*

*L153 Focal mechanisms of earthquakes

*This is added in line 107-108*

*L 167 remove 'fractures'

*Accepted.*

*L188: change into principal stress axes

*Accepted.*

*The authors wrote in the rebuttal : We use orientation and discontinuity type to define discontinuity sets, and group these into associations if they are mechanically consistent (i.e. they might have formed in the same stress field). We prefer to refrain from using relative timing to define discontinuity sets, as we believe cross-cutting and abutment relations are often ambiguous, and contain little meaning if the two discontinuities formed in the same stress field that might have prevailed for several millions of years.

I disagree with this statement, since I do think that for instance multiple discontinuity sets with similar orientations may have formed at different stages of the tectonic history. This is often observed in the field on the basis of relative chronology and has been confirmed recently by U-Pb geochronology on synkinematic calcite mineralization. I strongly believe that determining the relative chronology between individual fractures (mode I fractures or faults) using cross-cutting/abutting/reactivation criteria on a statistical basis remains an extremely valuable and necessary information for tectonic reconstructions. I therefore do believe that the definition of a fracture set should also include the relative timing.

*Maybe we didn't express ourselves well, we apologize for that. But we think that in the context of defining DAs, the relative timing between individual sets within such a DA is of secondary importance. The purpose of defining DAs is to reconstruct paleostress orientations that can be extrapolated to the subsurface, and the relative timing between sets within a DA is not going to reveal any new information regarding the paleostress orientation of that DA. Therefore, we did not include this criterion in the definition of discontinuity sets.*

*We did not mean to say that cross-cutting relationships have no value at all. In particular for the case of veins, where the cement of the youngest vein cross-cuts the cement of the older, it is a very clear indication for relative timing. This is what we used to determine relative timing between different DAs (but not between individual sets within a single DA).*

*To clarify this point in the manuscript, we have added a line (116-118):*

*'Relative timing based on cross-cutting and abutment relationships between individual sets within a discontinuity association are not considered in this study, as they do not reveal additional information on the orientation of the paleo principal stresses.'*

* The authors wrote in the rebuttal : It is more than likely that faults and discontinuities

related to the stress perturbation around the faults influence locally the flow characteristics of the reservoir, but that is outside the scope of the current study. Interestingly enough, we do find discontinuity associations that are similar in orientation to those observed in the other outcrops

(Parmelan, Jura) network close to the seismically active Vuache fault (see stations 32, 35 and 39 on figure 10). Therefore, we define them as part of the background network, that formed prior to tilting of the strata. So even though the complex history of this fault, with several reactivation phases, the regionally consistent background network is still present within several 10s of meters away from the Vuache fault.

I may partly agree, but this must be clearly stated in the manuscript, i.e., your answer to my earlier comment must be transfered into the text. It is of utmost importance in order to unambiguously show the interested reader that this point was not neglected a priori and that you are aware of stress perturbations in the vicinity of (strike-slip) faults (rotation of principal stress axes, variation of the stress ellipsoid shape ratio) and of the development of associations of secondary fractures formed in the perturbed stress field. I therefore invite you to add a few sentences on this point and to consider citing the papers dealing with directional stress perturbations in the vicinity of the major strike-slip faults in the area investigated (Jura) by Homberg et al., JSG 1997 and Homberg et al EPSL 2004.

I would add that according to these papers, the stress may be deviated over a 4-5 km wide area around the fault itself, and not 10s of meters as stated. It is clear from the map of Fig.10 that in sites 35 and 39, the compressional stress is rotated counterclockwise, which substantiates my comments and the need to add some cautionary sentences. In fact, it is because you accept a deviation of 30° in the orientation of your discontinuity sets that you can consider them to still reflect the regional background network despite obvious stress deviations. Again, this must be clearly stated.

*Accepted, we added this to discussion (line 302-308):*

*'Previous workers have demonstrated that in the Jura belt, northeast of the area of interest of this study, stress perturbations impacted the orientation of deformation structures around major left-lateral strike-slip faults (Pontarlier and Morez faults; Homberg et al., 1997, 2004). These faults are similarly oriented as the Vuache fault that bounds the Geneva Basin on the southwestern side. We document two stations in the vicinity of this fault (32 and 39, see figure 10), where the orientation of E1 is rotated 10◦ counterclockwise with respect to the average orientation of E1. It is possible that this small rotation is also related to a perturbation of the stress field around the Vuache fault. However, the relationship of E1 with the bedding indicates that it was formed prior to tilting of the strata, and therefore we consider it to be part of the background network.'*

\* The authors wrote in the rebuttal : As mentioned above, relative timing is of subordinate importance for defining discontinuity sets in our study, as we focus on associations of discontinuities and the stress field in which they are formed, rather than individual sets.

This looks like you choose to overlook the fact that similarly oriented discontinuity sets or associations of discontinuities (eg. mode I fractures) may have formed at different times as it is now demonstrated by U-Pb geochronology (eg, Zeboudj et al Geosciences, 2025), which would be documented otherwise if looking carefully and statistically at cross-cutting / abutting relationships or reactivation. May I recall that the chronology between E1 and E2 events is based on relative chronology criteria ? So even though you consider the relative timing between individual discontinuities within a given association to be of subordinate importance, you do use relative chronology between discontinuities belonging to different associations to define the sequence of development of the discontinuity associations, hence of the driving paleostress fields. So I find it weird and somewhat dangerous to depreciate the use of relative chronology in fracture studies as you seemingly do. I guess the reader would expect an even short note on that point.

*We thank the reviewer for highlighting this point and agree that a note should be added on this point (line 116-118). See our comment above for our explanation.*

\*The authors wrote in the rebuttal : We do not assume there is a regional consistent regional stress field, but the results show that in fact there is consistency with respect to orientation of σ1 that formed the discontinuity associations of the background network (see figure 10).Geochronology in the form of dating calcite veins is indeed a very interesting way to constrain timing of the formation of the background network. It will give a minimum age constraint on the forming of the discontinuities, as the timing of calcite precipitation does not necessarily coincide with the propagation of a fracture. On top of this, we don't know how much time is involved in the creation of the background network. If we consider that they are formed under sub-critical conditions, it might have taken 10s Ma.Therefore, it is expected that different ages will come out for the discontinuity association.

I cannot grasp the logic behind the last sentence regarding the present study. Calcite mineralization is undoubtedly syn-kinematic when associated from slickenfibers from striated mesoscale faults or when calcite has grown as fibers within opened veins. In case of blocky calcite textures, I agree that calcite mineralization may be delayed with respect

to opening at the scale of an individual vein, but one can safely consider that, for a vein set, calcite precipitation is coeval with vein opening over the time span corresponding to the fracturing event forming the vein set, and is thus syn-kinematic in a broad sense. This means that in the time interval required to form a discontinuity association you are interested in, calcite mineralizations can be considered as syn-kinematic for veins and faults (see Zeboudj et al Geosciences 2025) within the uncertainty of U-Pb calcite geochronology, so the range of consistent ages obtained for such given fracture association may give a hint to the duration of the related fracturing event/driving stress field. U-Pb calcite geochronology has also revealed that individual fractures as well as fracture associations may be wrongly interpreted to be coeval, hence formed in the same stress field, if the interpretation is done on the sole basis of their geometry and orientation, so going back and forth between absolute dating and 'classical' field-based approach (a fracture set defined on the basis of similar orientation, deformation mode AND relative chronology) is to date the more efficient way to meaningful tectonic reconstructions. I acknowledge absolute dating of fractures is beyond the scope of the study, but, again, a cautionary note would be very useful.

*We agree with the reviewer that absolute ages with U/Pb geochronology of calcite mineralization would be of great added value to the present study. It will indeed constrain the timing of events, and may reveal complex tectonic histories of the discontinuity network. The potential of geochronology was added in the manuscript in line 315-317.*

\* The authors wrote in the rebuttal : But in the current study, we don't have any absolute time constraints on the formation of DAs. That is why we focus on relative timing with respect to tilting/bending of the strata.

Yes (see above), but in fact you analyzed fractures with an Andersonian state of stress in mind, which is already an hypothesis in folded and faulted domains (again, see Tavani et al., 2025). As I wrote previously, the classical fold test must be preferred.

*All the data presented in figure 8 is backtilted with respect to the bedding, to investigate if the resulting princpial stress orientations are indeed indicative of an Andersonian state of stress.*

\* The authors wrote in the rebuttal : We added the references for stylolite as paleostress marker in the Introduction. Although it is a highly interesting topic, using stylolites to reconstruct magnitudes of paleostress is outside of the scope of the current manuscript, and therefore references are not included. However, for tectonic stylolites, this may be

challenging, as the assumption that σ2 and σ3 are of same magnitude is no longer valid, as for bed-parallel stylolites.

Yes, I agree, but in the case of tectonic stylolites you should be able to define whether the stylolites truly developed under a reverse or a strike-slip stress field, and therefore if they are mechanically consistent (or not) with the discontinuity association of interest to your tectonic reconstructions. Note that the references I provided pertain to tectonic stylolites only. I acknowledge this point goes beyond the scope of your manuscript.

*To use stylolites to distinguish between strike-slip and reverse regimes is an interesting point, and indeed would be a great complementary analysis for the work we present in this study.*

* The authors wrote in the rebuttal : In the Results section, when defining the regional events, we appreciate the fact that bed- perpendicular stylolites are indicative of a strike-slip or reverse regime.

Just to make things clear, you appreciate tectonic stylolites may be indicative of strike-slip or reverse regime because they are seemingly associated in the field with strike-slip or reverse faults, respectively. But this is some kind of circular reasoning. What I mean is that the real stress regime associated with stylolite development could be documented independently from the fracture associations by the approach mentioned above, the only to date to the best of my knowledge, providing a possible test on the validity of relating stylolites to the same association than conjugate faults and veins. However, I acknowledge that this point requires also the estimate of the depth of formation of the tectonic stylolite, and that the approach would need much more analyzing effort than collecting fracture data in poorly deformed sedimentary rocks.

*That clarifies the point. Indeed, such a study on stylolites would be a very nice addition to the presented work, and worth investigating in follow-up research.*

* The authors wrote in the rebuttal : We did not focus on the fold and/or fault that might be present in the well, because that is not the scope of the current manuscript. We focus on characterizing the background network based on outcrop observations. It is by no means the aim to present a 'complete' interpretation of the BHI that explains all features observed in the well.

Frankly speaking, you cannot simply brush this comment aside by claiming the occurrence of the fold is not the scope of the manuscript. Since you choose to show two

wells GeO-01 and GeO-02, you cannot overlook simply the fold occurrence in GeO-01.

The authors wrote in the rebuttal : in this study we focus on the background network and we prefer to turn the observation around: the same percentage of features in both wells can be related to the predicted background network. This underlines the regional character of the background network, or in other words, that this portion of the fractures may be extrapolated to the entire reservoir. As mentioned above, it was outside of the scope of the current study to investigate how the features that are not related to the background network compare between the two wells.

I may understand your point, but first in L 394 you wrote there was no fold ! Second, since you also wrote that a fold create some localized fracturing, the reader expects that crossing a fold will increase the number of fractures encountered in the well. How do you explain that the same percentage of features in both wells are related to the predicted background network, regardless of the folded or flat-lying character of strata ?

*The reviewer is correct that in line 394, we suggested that there is no fold/fault in the subsurface of the Geneva Basin, and this is not the case everywhere. The statement only applies for GEo-02, and the line is adjusted accordingly (now line 309-310). But also in Geo-02, almost half of the observed features fit within the background framework.*

*The question why we see the same percentage of features of background network in both wells is an interesting one. There are different explanations possible for this.*

*One possible explanation is that the background network may have a spatial variability, and clustering of background-discontinuities may occur. This could be the reason that the total background-related features in Geo-01 is higher than in Geo-02, but then the overprinting of the fold resulted in a similar percentage as in Geo-02.*

*Another reason could be the different available logs: for Geo-01, there is both OBI and ABI available, for Geo-02 only ABI (see line 238). It is possible that not all features present in well Geo-02 are observed due to the lack of OBI. Besides this, there might be a bias in the interpretation of Geo-01. We are currently re-evaluating multiple interpretations of this well, to assess the impact of bias on the final interpretation.*

*These considerations are added in the manuscript in line 360-370:*

*'The DA-methodology provides a prediction of the discontinuity type of up to 50% of the observed discontinuities in the two boreholes in the Geneva Basin, even though the resolution of the BHI is too low to determine discontinuity type, and there is no core*

*available to correlate the BHI with. The percentage of background-related discontinuities is similar in the two wells, whereas the total number of features is higher in GEo-01. A possible explanation is that there is only ABI-log available for GEo-02, and no OBI. It might be that not all discontinuities present in GEo-02 are visible on the ABI log, due to a lack of contrast in acoustic properties between host rock and discontinuity, resulting in a lower number of features picked. Another possibility is that the difference in number of background-related features is caused by the spatial variability within the background network. GEo-01 might have penetrated a denser part of the background network compared to GEo-02. After the emplacement of the background network, GEo-01 is affected by localized deformation (i.e. a fold, see figure 11), producing more discontinuities in this well. At present-day, the percentage of background-related discontinuities is then the same as in GEo-02.'*

*The authors wrote in the rebuttal : In this study, we document DAs in the field. It turns out there is a regional consistency in the orientation of DAs, regardless of the distance to regional faults such as the Vuache. We acknowledge that local stress perturbation in flat-lying strata may occur, but with the methodology presented in the current MS, they would not be documented, as they don't reveal a regional pattern.

I disagree. Local stress perturbations are likely documented but they are hidden behind the 30° deviation you accept for considering the discontinuity association to belong to the background pattern. Clearly, the stress is rotated counterclockwise on the map of figure 10 (sites 35 and 39). See also my earlier comment above. Again, a cautionary note is needed.

*Indeed, for E1, there is a counterclockwise rotation of 10° for S1, which falls within the deviation we consider acceptable for defining background-related discontinuities in the wells.  We added a cautionary note on the possibility of rotation of paleo principal stress axes due to perturbation around faults (see line 302-308).*

I hope that these new comments will help the authors clarify further their interpretations, introduce some cautionary notes about their choices and possible pitfalls of their approach and provide a more comprehensive description of the hypotheses behind their approach without omitting some specific points which would easily open room for criticism.
I am looking forward to seeing the revised manuscript published soon.